# Learning Time-shared Hidden Heterogeneity for Counterfactual Outcome Forecast

## Abstract

Forecasting counterfactual outcome in the longitudinal setting can be critical for many time-related applications. To solve this problem, the previous works propose to apply different sequence models including long short-term memory (LSTM) networks and transformers to model the relationship between the observed histories, treatments and outcomes, and apply various approaches to remove treatment selection bias. However, these methods neglect the hidden heterogeneity of outcome generation among samples induced by hidden factors which can bring hurdles to counterfactual outcome forecast. To alleviate this problem, we capture the hidden heterogeneity by recovering the hidden factors and incorporate it into the outcome prediction process. Specifically, we propose a Time-shared Heterogeneity Learning from Time Series (THLTS) method which infers the shared part of hidden factors characterizing the heterogeneity across time steps with the architecture of variational encoders (VAE). This method can be a flexible component and combined with arbitrary counterfactual outcome forecast method. Experimental results on (semi-)synthetic datasets demonstrate that combined with our method, the mainstream models can improve their performance.

## 1 Introduction

Decision-making problems are widely prevalent in many applications, such as healthcare (Huang & Ning, 2012) and marketing (Bottou et al., 2013). Therefore, it is of paramount importance to forecast the counterfactual outcome for different choice of treatments to assist decision. The gold standard for estimating the outcome of different treatments is conducting randomized controlled trials (RCTs) (Booth & Tannock, 2014), which randomly assign treatments to the samples. However, the high expense and time cost of RCTs (Kohavi & Longbotham, 2011) induce the people to instead learning from large amounts of observational data to fulfill this purpose. A lot of previous literature investigate the problem of counterfactual outcome forecast based on the observational dataset with different approaches to address the treatment selection bias (Hassanpour & Greiner, 2019; Assaad et al., 2021), such as treatment invariant representation learning (Johansson et al., 2016; Shalit et al., 2017; Tanimoto et al., 2021; Schwab et al., 2020; Yao et al., 2018; Zeng et al., 2020), sample re-weighting for adjusting distributions (Assaad et al., 2021; Hassanpour & Greiner, 2019; 2020; Johansson et al., 2018; Zou et al., 2020) and data imputation (Bica et al., 2020c; Yoon et al., 2018; Qian et al., 2021).

In many scenarios, the decision-making problems can be more complex and may span a long period of time. It is thus required to forecast the counterfactual outcome at different time steps instead of a single time. Since the size of historical covariate information is varied among the time steps, the methods developed under the static setting can not be directly applied to this setting. To bridge this gap induced by the longitudinal property of the task, some methods apply the sequence models, such as LSTM networks (Hochreiter & Schmidhuber, 1997) and transformers (Vaswani et al., 2017), to characterize the time-dependency between the histories of varying length and outcomes. Generally, a representation is extracted from the histories and play a substitute role of the raw confounder vectors in the static setting. The outcome prediction module forecasts the counterfactual outcome with the learned representation and the counterfactual treatment. Based on this design, the technologies for removing treatment selection bias, can be integrated to achieve more accurate outcome forecast in the setting of time-series.

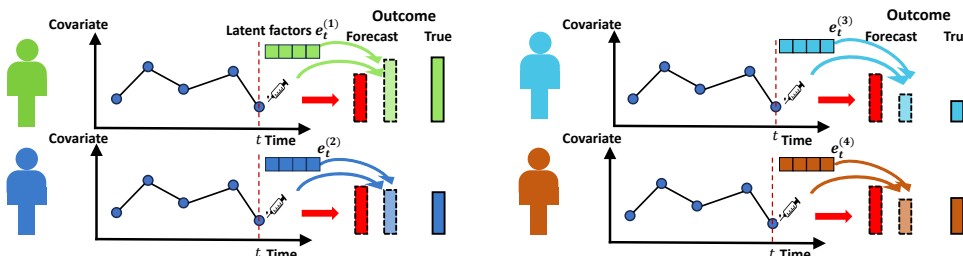

Figure 1: The diagram of counterfactual outcome forecast with hidden heterogeneity. Although the observed histories of the four individuals are same, the true counterfactual outcomes of them are in significantly distinct due to the latent factors. Predicting based on solely the same history results in the same prediction which brings hurdle to the performance. By uncovering latent factors capturing the hidden heterogeneity, we can achieve more precious prediction result.

The paradigm above attributes the heterogeneity of outcome generation among samples to the observed variables of histories in dataset. This hypothesizes that the outcome-related factors are all recorded in the histories. However, in many scenarios, this prerequisite does not hold. There may exist extra factors unrecorded by the observed histories, and can also affect outcome. Thereby, forecasting heterogeneous outcome conditional on solely the observed histories may neglect the outcome variation among samples (i.e. heterogeneity of outcome generation) of the same observed variables. Consequently, the neglect of this outcome variation can lead the prediction to be a coarse approximation of individual outcome and deteriorate the forecast performance (Zou et al., 2023). We call this problem as hidden heterogeneity and visually demonstrate it in Figure 1.

To resolve the problem, we try to address the hidden heterogeneity problem by uncovering the hidden factors and capturing the extra outcome variation. Due to the limited supervision information (i.e. outcome variable is of few-dimension) in this problem, the solution by aggressively learning latent factors for all samples and time steps is excessively flexible and undergoes sub-optimal performance, which is empirically presented in the experiments. To mitigate this circumstance, we propose a novel Time-shared Heterogeneity Learning from Time Series (THLTS) method, which instead learn the shared part of latent factors across time steps for each sample. While sacrificing the flexibility in modeling time-varying dynamics of latent factors, this design is targeting the pursuit of forecast performance like a regularizer.

For practical implementations, we resort to variational autoencoders (VAEs) (Kingma & Welling, 2014; Rezende et al., 2014) to model the joint distribution of the outcome and the time-shared latent factors given the observed history and treatments. Due to the longitudinal property of the problem, we extend the model architecture to support the inference of latent factors with the varying prior, which are updated according to the successive observation over time. With the encoder component, we can easily infer the latent factors. By incorporating the learned factors into outcome prediction module, we can forecast the individual outcome more preciously compared to the same model-backbone ignoring the hidden heterogeneity. Theoretical analysis validate the rationality of our proposed strategy to learn time-shared latent factors. We conduct extensive experiments on the synthetic datasets and semi-synthetic datasets, where the results reveal the effectiveness of our method.

The main contribution of our paper can be summarized as following:

- This paper investigate counterfactual outcome forecast with the existence of hidden heterogeneity. To the best of our knowledge, this is the pioneer work tailored to improving the off-the-shelf counterfactual forecast model by addressing the hidden heterogeneity problem.

- We expose the insightful idea that learning the shared part of latent factors over time steps and propose a novel Time-shared Heterogeneity Learning from Time Series (THLTS) method.

- Extensive experimental results demonstrate that our method acts like a flexible component and is beneficial for the mainstream counterfactual outcome forecast models when be integrated into them.

## 2 RELATED WORKS

We respectively review the related literature under both static setting and longitudinal setting.

### 2.1 COUNTERFACTUAL OUTCOME PREDICTION IN THE STATIC SETTING

There have been a large amount of works devoted to counterfactual outcome forecast in the static setting. The main challenge is the treatment selection bias manifested as the dependency of treatment assignment on the observed confounders. To overcome this challenge, some papers (Johansson et al., 2016; Shalit et al., 2017; Tanimoto et al., 2021; Schwab et al., 2020; Yao et al., 2018; Zeng et al., 2020) borrow the idea of domain adaptation (Tzeng et al., 2014; Ganin & Lempitsky, 2015) to learn the treatment invariant representation of confounders and predict counterfactual outcome by taking the representation as input. Since over-forcing the independency may bring hurdles to prediction performance (Assaad et al., 2021), some other methods (Assaad et al., 2021; Hassanpour & Greiner, 2019; 2020; Johansson et al., 2018; Zou et al., 2020) re-weight samples to adjust the joint distribution of confounders and treatments, and train the counterfactual predictive model on the re-weighted dataset. Moreover, there are also some works perform data augmentation to imputes the counterfactual outcome for the observational samples (Bica et al., 2020c; Yoon et al., 2018; Qian et al., 2021). When faced with unobserved confounders, the prediction result may suffer from severe confounding bias. The previous literature resort to extra tools, such as instrumental variables (IVs) and negative controls (Hartford et al., 2017; Heckman, 1997; Wu et al., 2022) to overcome it. Since the assumptions on these tools are restricted, other methods (Louizos et al., 2017; Wang & Blei, 2019; Zou et al., 2023) try to recover the information of unobserved confounders according to the knowledge of data generation process.

### 2.2 COUNTERFACTUAL OUTCOME FORECAST IN TIME SERIES

In many applications, it is required to forecast counterfactual outcome for a period of time. Under these scenarios, the outcome of a specific time is determined by not only the observations at the present time but also the whole histories. To handle the information of long history, Robins et al. (2000) conduct linear/logistic regression on the truncated history to predict the outcome. Due to the potential complex dependency of outcome on covariates and treatments, previous works (Lim et al., 2018; Bica et al., 2020b; Li et al., 2020; Melnychuk et al., 2022; Hatt & Feuerriegel, 2021; Bica et al., 2020a; Kuzmanovic et al., 2021) propose to utilize sequence models (Chung et al., 2014; Vaswani et al., 2017; Hochreiter & Schmidhuber, 1997) to transform the histories of varying length to representations of fixed size and forecast outcome based on it. Lim et al. (2018) utilize long short-term memory (LSTM) networks to encode the histories and re-weights the samples by the estimated propensity scores for removing selection bias. Alternatively, Bica et al. (2020b) resort to impose treatment invariant representation regularizer on the represention learned by LSTM networks to remove treatment selection bias. Since the ability of LSTM networks to model complex and long dependencies in time is limited, Melnychuk et al. (2022) build the model based on transformers (Vaswani et al., 2017) and propose a novel counterfactual domain confusion (CDC) loss for training the models. Bouchattaoui et al. (2023) is a similar work to ours. However, this work designed a fixed model architecture which can only deal with the problem setting of binary treatment. By contrast, our proposed THLTS method can be viewed as a flexible component which is able to be combined with various model backbones of counterfactual outcome forecast and deal with more complex treatment scenarios. Moreover, this paper also consider the setting of time-varying latent factors and delivers the insightful strategy of learning shared part of latent factors across time steps to deal with it.

## 3 PROBLEM FORMULATIONS AND STATEMENTS

We define $\mathbf{X} \in \mathcal{X} \subset \mathbb{R}^{d_x}$ as the observed covariate vector which records the individual information of each sample, $\mathbf{A} \in \mathcal{A}$ as the treatment and $\mathbf{Y} \in \mathbb{R}$ as outcome. Therefore, the observational datasets is denoted as $\mathcal{D} = \{\{\mathbf{x}_t^{(i)}, \mathbf{a}_t^{(i)}, \mathbf{y}_t^{(i)}\}_{t=1}^{T^{(i)}}\}_{i=1}^{n}$, where the superscripts and subscripts refer to the sample indexs and time indexs respectively. For the sake of description, we abbreviate the history of individuals by $\mathbf{X}_{t:t+\tau}^{(i)} = (\mathbf{x}_t^{(i)}, \mathbf{x}_{t+1}^{(i)}, \mathbf{x}_{t+2}^{(i)}, ..., \mathbf{x}_{t+\tau}^{(i)})$, $\mathbf{A}_{t:t+\tau}^{(i)} = (\mathbf{a}_t^{(i)}, \mathbf{a}_{t+1}^{(i)}, \mathbf{a}_{t+2}^{(i)}, ..., \mathbf{a}_{t+\tau}^{(i)})$, $\mathbf{Y}_{t:t+\tau}^{(i)} = (\mathbf{y}_t^{(i)}, \mathbf{y}_{t+1}^{(i)}, \mathbf{y}_{t+2}^{(i)}, ..., \mathbf{y}_{t+\tau}^{(i)})$ and $\mathbf{H}_t^{(i)} = \{\mathbf{X}_{1:t}^{(i)}, \mathbf{A}_{1:t-1}^{(i)}, \mathbf{Y}_{1:t-1}^{(i)}\}$.

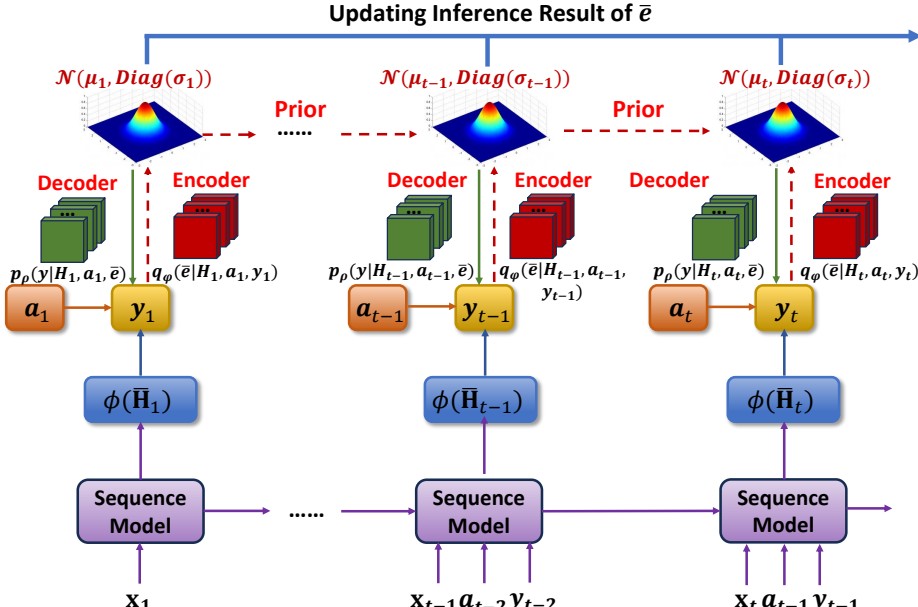

Figure 2: The diagram of time-shared heterogeneity learning framework.

The ultimate target is the outcome function at individual-level $\mathbf{Y}_{t+\tau}^{(i)}(\mathbf{A}_{t:t+\tau}^c)$, which refers to the counterfactual outcome of the $i^{th}$ sample at the $(t+\tau)^{th}$ time step under the counterfactual treatments $\mathbf{A}_{t:t+\tau}^c$. Many methods have been proposed in the previous literature to estimate the expected outcome given counterfactual treatments and histories, that is $\mathbb{E}[\mathbf{Y}_{t+\tau}|do(\mathbf{A}_{t:t+\tau} = \mathbf{A}_{t:t+\tau}^c), \mathbf{H}_t^{(i)}]$. This estimator attributes the heterogeneity of outcome generation to the observed historical information. However, in many scenarios, some factors that can also affect the outcomes may be not recorded, which we denote by $\mathbf{e}_t^{(i)} \in \mathcal{E} \subset \mathbb{R}^{d_e}$. This potentially results in extra hidden heterogeneity of outcome among different samples and time steps (Zou et al., 2023), which cause that the expected outcome conditional on histories is not equal to the true individual outcome. Formally, that is

$$\mathbb{E}[\mathbf{Y}_{t+\tau}^{(i)}(\mathbf{A}_{t:t+\tau}^c)] \neq \mathbb{E}[\mathbf{y}_{t+\tau}|do(\mathbf{A}_{t:t+\tau} = \mathbf{A}_{t:t+\tau}^c), \mathbf{H}_t^{(i)}].$$

To bridge this gap, we propose to uncover the hidden factors related to outcomes and augment the forecast module with the learned factors. By capturing the hidden heterogeneity, we can forecast the counterfactual outcome more preciously for the individuals. In this paper, we assume the latent factors do not affect treatment assignment, and therefore the standard assumptions (Rosenbaum & Rubin, 1983) in causal inference hold. The identification condition of counterfactual outcome are satisfied. We mainly consider the 1-step forecast problem (i.e. $\tau = 1$). The problem of multi-step forecast is left to future work.

## 4 THLTS: THE PROPOSED METHOD

In this section, we present the details of our proposed Time-shared Heterogeneity Learning from Time Series (THLTS) algorithm. It is a flexible component that can be combined with the different models and subsequently improve their performance.

### 4.1 PRELIMINARY FOR CAPTURING HIDDEN HETEROGENEITY

The mainstream models predict counterfactual outcome based on the heterogeneity informed by the representation encoding the histories. Formally, the prediction can be expressed by $h(\phi(\mathbf{H}_t^{(i)}), \mathbf{a}_t^c)$, where $\phi(\cdot)$ is the representation function built upon sequence models, and $h$ is the predictive model.

However, hidden heterogeneity of outcome generation beyond the histories can still exist among different samples and time steps, which is caused by hidden outcome-related factors $\mathbf{e}_t^{(i)}$. It has

been verified in previous literature (Zou et al., 2023) that by augmenting the prediction model input with the extra hidden factors, that is $g(\phi(\mathbf{H}_t^{(i)}), \mathbf{e}_t^{(i)}, \mathbf{a}_t^c)$, we can approximate the true individual outcome more closer. There have been large amount of works in reinforcement learning investigating the problem of partially observed markov decision process (POMDP) and trying to recover latent states(Lee et al., 2020; Lei et al., 2022; Igl et al., 2018) across time steps. The supervision information for time-varying latent variable inference (i.e. the high-dimensional observation) is usually sufficient in POMDP. Conversely, the supervision information (i.e. outcome variable) is much limited in our problem, since it is only one-dimensional. Therefore, the models built for capturing the time-varying latent factors is excessively flexible and may result in sub-optimal performance. This will be presented thoroughly in the section of experiments. To mitigate this circumstance, we give an subtle method to learn the shared part of time-varying latent factors. This is designed for computational efficiency while trading off the flexibility of models.

### 4.2 LEARNING TIME-SHARED LATENT FACTORS

Inspired by the analysis above, we attempt to build up a framework that infer the shared part $\bar{\mathbf{e}}$ of time-varying latent factors $\{\mathbf{e}_t\}_{t=1}^T$, and learn the forecast model $g$ taking the histories, time-shared latent factors and treatments as input. Through theoretical analysis below, we determine to learn the mean value of factor factors across time steps as the time-shared latent factors.

**Proposition 4.1.** *Assuming the function of prediction model $g$ is $\beta$-Lipschitz on $\mathbf{e}$, formally $|g(\phi(\mathbf{H}), \mathbf{e}, \mathbf{a}) - g(\phi(\mathbf{H}), \mathbf{e}', \mathbf{a})| \leq \beta \cdot ||\mathbf{e} - \mathbf{e}'||_2$, then the total increased error across time induced by substitute time-varying $\mathbf{e}_t$ with constant $\bar{\mathbf{e}}$ can be characterized as following:*

$$\sum_{t=1}^T (g(\phi(\mathbf{H}_t), \bar{\mathbf{e}}, \mathbf{a}_t^c) - \mathbf{Y}_t(\mathbf{a}_t^c))^2 \leq 2 \sum_{t=1}^T (g(\phi(\mathbf{H}_t), \mathbf{e}_t, \mathbf{a}_t^c) - \mathbf{Y}_t(\mathbf{a}_t^c))^2 + 2\beta^2 \cdot \sum_{t=1}^T ||\mathbf{e}_t - \bar{\mathbf{e}}||_2^2 \quad (1)$$

We observe that when the latent factors is substituted as the mean value across time steps, the upper bound in the r.h.s of Equation 1 reaches the minimal value. Formally, the substituted latent factors is

$$\bar{\mathbf{e}} = \frac{1}{T} \sum_{t=1}^T \mathbf{e}_t \quad (2)$$

We can observe from Equation 2 that when the latent factors are generated from a distribution with constant expectation, our pursuit becomes to infer the expectation for each sample. This is a first-order statistic of latent factors and can be learned with variational inference technology (Kingma & Welling, 2014; Rezende et al., 2014).

**Proposition 4.2.** *We assume the following conditions are satisfied:*

1. *The latent factors are generated by $\mathbf{e}_t = \bar{\mathbf{e}} + \eta_t$, where $\eta_t \sim p(\eta)$ is a noise term with zero mean.*

2. *The outcome distribution $p(\mathbf{y}_t|\mathbf{H}_t, \mathbf{a}, \bar{\mathbf{e}}) = \int_\eta p(\mathbf{y}_t|\mathbf{H}_t, \mathbf{a}, \mathbf{e} = \bar{\mathbf{e}} + \eta) \cdot p(\eta)d\eta$ is in the function family of decoder $p_\rho(\mathbf{y}_t|\mathbf{H}_t, \mathbf{a}, \bar{\mathbf{e}})$.*

3. *The posterior distribution $p(\bar{\mathbf{e}}|\mathbf{H}_t, \mathbf{a}_t, \mathbf{y}_t)$ is in the function family of encoder $q_\varphi(\bar{\mathbf{e}}|\mathbf{H}_t, \mathbf{a}_t, \mathbf{y}_t)$.*

*Then there is an optimal solution for maximizing the evidence lower bound (ELBO) of variational autoencoders, which characterizes the underlying data generation process:*

$$q_\varphi(\bar{\mathbf{e}}|\mathbf{H}_t, \mathbf{a}_t, \mathbf{y}_t) = p(\bar{\mathbf{e}}|\mathbf{H}_t, \mathbf{a}_t, \mathbf{y}_t), \quad p_\rho(\mathbf{y}|\mathbf{H}_t, \mathbf{a}, \bar{\mathbf{e}}) = p(\mathbf{y}|\mathbf{H}_t, \mathbf{a}, \bar{\mathbf{e}}), \quad p_\rho(\bar{\mathbf{e}}|\mathbf{H}_t) = p(\bar{\mathbf{e}}|\mathbf{H}_t)$$

*where the ELBO is defined as:*

$$\sum_{i=1}^n \sum_{t=1}^{T^{(i)}} \left( \mathbb{E}_{\bar{\mathbf{e}} \sim q_\varphi(\bar{\mathbf{e}}|\mathbf{H}_t^{(i)}, \mathbf{a}_t^{(i)}, \mathbf{y}_t^{(i)})} \left[ \log p_\rho(\mathbf{y}_t^{(i)}|\mathbf{H}_t^{(i)}, \mathbf{a}_t^{(i)}, \bar{\mathbf{e}}) \right] + D_{KL}(p_\rho(\bar{\mathbf{e}}|\mathbf{H}_t^{(i)})|q_\varphi(\bar{\mathbf{e}}|\mathbf{H}_t^{(i)}, \mathbf{a}_t^{(i)}, \mathbf{y}_t^{(i)})) \right)$$

Therefore, the mean of latent factors can be learned with the architecture of variational autoencoders (VAEs) by assuming the learned latent factors keeps constant across time and maximizing the ELBO. The detailed background information of VAEs can be found in the section F of Appendix.

Although the learned time-shared latent factors can not exactly capture the time-varying process of latent factors, it still can facilitate more precious counterfactual outcome forecast with trade-off between model flexibility and computation efficiency. This will be empirically validated in the section of experimental results. The detailed proof of the propositions above can be found in Section C of Appendix.

### 4.3 IMPLEMENTATIONS

In this paper, we choose the architecture of variational autoencoders (VAEs) to infer time-shared latent factors $\bar{\mathbf{e}}^{(i)}$ for each sample. The instantiated model architecture consists of three parts, which are inference result memory, encoders and forecast models respectively. The overall framework is demonstrated in the Figure 2. We successively introduce the components in this subsection.

**Inference Result Memory** Since the stochastic nature of the data generation process, the time-shared latent factors can not be inferred with a deterministic manner. Hence, our inference result is expressed by a distributional estimation rather than point-wise estimation. Inspired by the common practice in variational inference, the estimated distribution is characterized by a gaussian distribution with the mean vector $\mu_t^{(i)}$ and variance vector $\sigma_t^{(i)}$. When there is no observation of treatment outcome at the initial time step, we set the initial distributional inference of latent factors as standard gaussian distribution $\mathcal{N}(\mathbf{0}, \mathbf{I}_e)$. Formally, the vector of mean and variance is set as $\mu_0^{(i)} = \mathbf{0}_e, \sigma_0^{(i)} = \mathbf{1}_e$. We keep record of the inferred distribution information for each sample. When new observations of treatment outcome arrive, we can obtain the new inference result (i.e. the mean vector $\mu_i$ and variance vector $\sigma_i$) and update the distribution record for the corresponding sample.

**Latent Factor Encoders** When the outcome $\mathbf{y}_{t+1}^{(i)}$ of new treatment $\mathbf{a}_{t+1}^{(i)}$ is observed, we can update the inference result of latent factors based on the new observation. According to Bayes' theorem (Davies, 1988), the posterior is determined by the likelihood and prior distribution. Therefore, the input of our encoders includes not only the information of observations (i.e. new outcome $\mathbf{y}_{t+1}^{(i)}$, new treatments $\mathbf{a}_{t+1}^{(i)}$ and representation of histories $\phi(\mathbf{H}_{t+1}^{(i)})$), but also the mean $\mu^{pr}$ and variance $\sigma^{pr}$ of prior distribution $p_\rho(\bar{\mathbf{e}}|\mathbf{H}_{t+1})$. Specifically, the encoder $q_\varphi(\cdot)$ represents a gaussian distribution characterized by two deep neural networks $f_\varphi^\mu(\cdot)$ and $f_\varphi^\sigma(\cdot)$ as following:

$$q_\varphi(\bar{\mathbf{e}}|\mathbf{H}_{t+1}^{(i)}, \mathbf{a}_{t+1}^{(i)}, \mathbf{y}_{t+1}^{(i)}) = \mathcal{N}(f_\varphi^\mu(\mathbf{y}_{t+1}^{(i)}, \mathbf{a}_{t+1}^{(i)}, \phi(\mathbf{H}_{t+1}^{(i)}), \mu^{pr}, \sigma^{pr}),$$
$$\text{Diag}(f_\varphi^\sigma(\mathbf{y}_{t+1}^{(i)}, \mathbf{a}_{t+1}^{(i)}, \phi(\mathbf{H}_{t+1}^{(i)}), \mu^{pr}, \sigma^{pr})^2)), \tag{3}$$

where $\varphi$ is the parameters of deep neural networks and $\phi(\cdot)$ is implemented by sequence model. The specification of $\mu^{pr}$ and $\sigma^{pr}$ will be demonstrated in the part of training process.

**Forecast Model** The forecast model $g_\rho(\cdot)$ output the counterfactual outcome based on the counterfactual treatments $\mathbf{a}^c$, the representation of histories $\phi(\mathbf{H}_{t+1}^{(i)})$ and inferred latent factors $\bar{\mathbf{e}}_i$. By exploiting the hidden heterogeneity encoded in $\bar{\mathbf{e}}_i$, we can forecast counterfactual outcome more accurately. The model $g_\rho(\cdot)$ can act as the key component of the decoder in VAEs and therefore be trained with the encoder together by the technology of variational inference.

**Training Process** We train the model components above by decomposing the histories into several time steps. According to the Bayes' Theorem, the obtained posterior distribution can be viewed as the prior distribution of the next time step. Therefore, the obtained $\mu_{t-1}^{(i)}$ and $\sigma_{t-1}^{(i)}$ can substitute the role of $\mu^{pr}$ and $\sigma^{pr}$ for the $t^{th}$ time step. Specifically, the objective function of the $i^{th}$ sample for training at the $t^{th}$ time step is:

$$\mathcal{L}_t^{(i)} = \mathbb{E}_{\bar{\mathbf{e}} \sim q_\varphi(\bar{\mathbf{e}}|\mathbf{H}_t^{(i)}, \mathbf{a}_t^{(i)}, \mathbf{y}_t^{(i)})} \Big[ \log p_\rho(\mathbf{y}_t^{(i)}|\mathbf{H}_t^{(i)}, \mathbf{a}_t^{(i)}, \bar{\mathbf{e}}) -$$
$$D_{KL} \Big( q_\varphi(\bar{\mathbf{e}}|\mathbf{H}_t^{(i)}, \mathbf{a}_t^{(i)}, \mathbf{y}_t^{(i)}) \Big| \mathcal{N} \Big( \mu_{t-1}^{(i)}, \text{Diag}((\sigma_{t-1}^{(i)})^2) \Big) \Big) \Big].$$

The decoded outcome distribution $p_\rho(\mathbf{y}_t^{(i)}|\mathbf{H}_t^{(i)}, \mathbf{a}_t^{(i)}, \bar{\mathbf{e}})$ is determined by the forecast model, formally $p_\rho(\mathbf{y}_t^{(i)}|\mathbf{H}_t^{(i)}, \mathbf{a}_t^{(i)}, \bar{\mathbf{e}}) = \mathcal{N}(g_\rho(\mathbf{a}_t^{(i)}, \phi(\mathbf{H}_t^{(i)}), \bar{\mathbf{e}}), (\sigma_y)^2)$, where $\sigma_y$ is set as hyper-parameter.

We model the posterior distribution of time-shared latent factors as the gaussian distribution $\mathcal{N}(\mu_t^{(i)}, \text{Diag}((\sigma_t^{(i)})^2))$ obtained from the encoder. To be specific, the vectors of mean and variance are learned as following:

$$
\begin{aligned}
\mu_t^{(i)} &= f_\varphi^\mu(\mathbf{y}_t^{(i)}, \mathbf{a}_t^{(i)}, \phi(\mathbf{H}_t^{(i)}), \mu_{t-1}^{(i)}, \sigma_{t-1}^{(i)}), \\
\sigma_t^{(i)} &= f_\varphi^\sigma(\mathbf{y}_t^{(i)}, \mathbf{a}_t^{(i)}, \phi(\mathbf{H}_t^{(i)}), \mu_{t-1}^{(i)}, \sigma_{t-1}^{(i)}).
\end{aligned}
\tag{4}
$$

For the initial time step, the prior distribution is set as $\mu_0^{(i)} = \mathbf{0}_e, \sigma_0^{(i)} = \mathbf{1}_e$. Finally, inspired by the analysis in Proposition 4.2, we define the objective function for training the model as the sum of $\mathcal{L}_t^{(i)}$ among the samples and time steps:

$$
\mathcal{L} = \sum_{i=1}^n \sum_{t=1}^{T^{(i)}} \mathcal{L}_t^{(i)}.
$$

**Forecast Process** When the model has been trained, we can forecast counterfactual outcome from the model $g_\rho(\cdot)$ based on simultaneously histories, treatments and the inferred latent factors. For the individual with observed histories $\mathbf{H}_{t+1}$, counterfactual treatment $\mathbf{a}_{t+1}^c$, the counterfactual outcome is estimated with the following two steps.

At the first step, the posterior distribution of time-shared latent factors $\bar{\mathbf{e}}$ is obtained by successively feeding the elements of histories $\{(\mathbf{y}_j, \mathbf{a}_j, \phi(\mathbf{H}_j))\}_{j=1}^t$ into the encoder $q_\varphi(\cdot)$ and updating the parameters of inferred distribution $\mu_j$ and $\sigma_j$ until $j = t$ according to Equation 4. The resulting distribution of latent factors is $\mathcal{N}(\mu_t, \text{Diag}((\sigma_t)^2))$.

Secondly, we repeatedly sample latent factors for $m$ times and empirically estimate the expectation $\mathbb{E}[g_\rho(\mathbf{a}_{t+1}^c, \phi(\mathbf{H}_{t+1}), \bar{\mathbf{e}})]$ as the forecast result. Formally, the estimated result is as following:

$$
\hat{\mathbf{y}}_{t+1}(\mathbf{a}_{t+1}^c) = \frac{1}{m} \sum_{j=1}^m g_\rho(\mathbf{a}_{t+1}^c, \phi(\mathbf{H}_{t+1}), \bar{\mathbf{e}}^j),
\tag{5}
$$

$$
\bar{\mathbf{e}}^j \sim \mathcal{N}(\mu_t, \text{Diag}((\sigma_t)^2)), \quad 1 \le j \le m
$$

The pseudo-code of whole algorithm can be found in Algorithm 1 of Appendix. It is noteworthy that our model and forecast process is agnostic to the architecture of sequence model. Therefore, our method can be viewed as a flexible model component. When plugged into any off-the-shell counterfactual forecast model, it can significantly enhance the forecast performance. This claim will be substantiated in the section of experiments.

# 5 EXPERIMENTS

In this section, we first evaluate the effectiveness of THLTS by conducting various experiments using synthetic dataset. To further verify the effectiveness of THLTS in real scenarios, we leverage a real-world medical dataset MIMIC-III (Johnson et al., 2016; Wang et al., 2020) to construct a semi-synthetic dataset.

## 5.1 EXPERIMENTAL SETUP

We validate the effectiveness of our THLTS method by combining it with the representative instances of the advanced counterfactual forecast models in time series, that are long-short term memory (LSTM) architecture with IPS-Reweighting (RMSN) (Lim et al., 2018), treatment invariant representation (CRN) (Bica et al., 2020b) and Transformer architecture with treatment invariant representation (Causal Transformer) (Melnychuk et al., 2022) respectively. We also compare the forecast performance with several other methods, that are G-net (Li et al., 2020) and MSM (Robins et al., 2000). Furthermore, to justify the rationality of learning shared part of latent factors, we introduce a new

Table 1: Experimental results for synthetic dataset under the setting of varying strength of latent factors, the average RMSE $\pm$ standard deviation is recorded for 10 repeated experiments.

| | Fix the sequence length $d = 20$, varying the latent factor strength $\sigma_e$ | | | | |
|---|---|---|---|---|---|
| $\sigma_e$ | $\sigma_e = 0.5$ | $\sigma_e = 1.0$ | $\sigma_e = 1.5$ | $\sigma_e = 2.0$ | $\sigma_e = 2.5$ |
| MSM | $0.657 \pm 0.000$ | $0.831 \pm 0.000$ | $1.134 \pm 0.000$ | $1.375 \pm 0.000$ | $1.626 \pm 0.000$ |
| G-Net | $0.542 \pm 0.030$ | $0.601 \pm 0.021$ | $0.719 \pm 0.026$ | $0.851 \pm 0.050$ | $0.958 \pm 0.041$ |
| RMSN | $0.379 \pm 0.013$ | $0.525 \pm 0.019$ | $0.658 \pm 0.019$ | $0.797 \pm 0.042$ | $0.924 \pm 0.032$ |
| RMSN-THLTS$^{(v)}$ | $0.338 \pm 0.010$ | $0.462 \pm 0.009$ | $0.573 \pm 0.017$ | $0.689 \pm 0.017$ | $0.799 \pm 0.017$ |
| RMSN-THLTS | $0.328 \pm 0.007$ | $0.441 \pm 0.009$ | $0.557 \pm 0.011$ | $0.673 \pm 0.021$ | $0.783 \pm 0.020$ |
| CRN | $0.302 \pm 0.008$ | $0.437 \pm 0.009$ | $0.551 \pm 0.017$ | $0.665 \pm 0.020$ | $0.777 \pm 0.019$ |
| CRN-THLTS$^{(v)}$ | $0.278 \pm 0.004$ | $0.392 \pm 0.007$ | $0.495 \pm 0.014$ | $0.604 \pm 0.016$ | $0.722 \pm 0.016$ |
| CRN-THLTS | $\mathbf{0.266 \pm 0.005}$ | $\mathbf{0.364 \pm 0.007}$ | $\mathbf{0.463 \pm 0.006}$ | $\mathbf{0.567 \pm 0.003}$ | $\mathbf{0.677 \pm 0.004}$ |
| CT | $0.366 \pm 0.012$ | $0.428 \pm 0.005$ | $0.521 \pm 0.005$ | $0.620 \pm 0.006$ | $0.840 \pm 0.007$ |
| CT-THLTS$^{(v)}$ | $0.471 \pm 0.011$ | $0.609 \pm 0.002$ | $0.738 \pm 0.014$ | $0.903 \pm 0.020$ | $1.096 \pm 0.039$ |
| CT-THLTS | $0.296 \pm 0.009$ | $0.396 \pm 0.010$ | $0.489 \pm 0.005$ | $0.599 \pm 0.009$ | $0.712 \pm 0.014$ |

baseline method for comparison, which is denoted as THLTS$^{(v)}$. It leverages a linear layer $\psi(\cdot)$ to replace the priors of latent factors $\mu^{pr}$ and $\sigma^{pr}$ by $\psi(\mu_{t-1}^{(i)})$ and $\psi(\sigma_{t-1}^{(i)})$ respectively to encourage the model to capture the temporal variability.

## 5.2 SYNTHETIC EXPERIMENT

**Dataset.** We simulated various synthetic dataset under different setting to evaluate the effectiveness of THLTS. For each sample, we sample the covariates $\mathbf{x}_i$ with dimension $d_x$ at the time step $t = 0$ independently from Gaussian Distribution:

$$\mathbf{x}_{t=0}^{(i)} \sim \mathcal{N}(0, \mathbf{I}_{d_x}). \tag{6}$$

We set the dimension of context $d_x$ to be 10 across all synthetic settings. The covariates of each sample at the $t^{th}$ time step depends on the previous covariates and treatment:

$$\mathbf{x}_t^{(i)} = \frac{1}{ws} \sum_{j=t-ws}^{t-1} \mathbf{x}_j^{(i)} + \mathbf{a}_{t-1}^{(i)} \cdot A + \mathcal{N}(0, 0.3^2 I), \tag{7}$$

where $ws$ denotes the window size that determines the influence of previous covariates, and $A \in R^{p \times 1}$ is the constant vector sampled from $\mathcal{N}(0, \mathbf{I}_p)$. The treatment assignment is confounded by the current covariates:

$$a_i^t \sim Bernoulli(\beta^T x_t^{(i)}), \tag{8}$$

where $\beta \in \mathbb{R}^{p \times 1}$ is a parameter vector sampled from Gaussian Distribution $\mathcal{N}(0, \mathbf{I}_p)$. The outcome of each sample is generated as following:

$$\mathbf{y}_t^{(i)} = B \cdot [\mathbf{x}_{t-ws+1}^{(i)}, \mathbf{x}_{t-ws+2}^{(i)}, ..., \mathbf{x}_t^{(i)}] \cdot C + CE(t) \cdot \mathbf{e}_t^{(i)} + \mathcal{N}(0, 0.5^2), 1 \leq t \leq T^{(i)} \tag{9}$$

where $B \in \mathbb{R}^{ws \times 1}, C \in \mathbb{R}^{p \times 1}$ are the weight vectors with each element sampled from $\mathcal{N}(0, 1)$, $CE(t) = \frac{CE(t-1)}{2} + \mathbf{a}_t^{(i)}$ is the time-decaying treatment effect at the $t^{th}$ time step. In the synthetic dataset, the time horizon of each sample is set to be constant $T^{(i)} = d, 1 \leq i \leq n$. The variable $\mathbf{e}_t^{(i)}$ is unrecorded in the histories $\mathbf{H}_t$ and encodes the hidden heterogeneity. To verify the necessity of dealing with hidden heterogeneity, we firstly consider the setting of static latent factors across time steps. The latent factors are generated with the following two steps:

$$\bar{\mathbf{e}}^{(i)} \sim \mathcal{N}(0, \sigma_e^2), \quad \mathbf{e}_t^{(i)} = \bar{\mathbf{e}}^{(i)}, \quad 1 \leq t \leq d \tag{10}$$

where $\sigma^e$ is a constant that controls the strength of latent factors.

**Results.** For each experimental setting, we have conducted repeated experiments for 10 times, and we record the average RMSE and standard deviation for each method.

Table 2: Synthetic experimental results under different trajectory horizon $d$, the average RMSE $\pm$ standard deviation is recorded for 10 repeated experiments.

| Fix the hidden factor strength $\sigma_e$ = 1.5, varying the sequence length $d$ | | | | | |
|---|---|---|---|---|---|
| $d$ | $d = 10$ | $d = 15$ | $d = 20$ | $d = 25$ | $d = 30$ |
| MSM | $1.178 \pm 0.000$ | $1.123 \pm 0.000$ | $1.134 \pm 0.000$ | $1.086 \pm 0.000$ | $1.120 \pm 0.000$ |
| G-Net | $0.758 \pm 0.009$ | $0.696 \pm 0.017$ | $0.719 \pm 0.026$ | $0.731 \pm 0.034$ | $0.796 \pm 0.022$ |
| RMSN | $0.729 \pm 0.012$ | $0.650 \pm 0.028$ | $0.658 \pm 0.019$ | $0.660 \pm 0.043$ | $0.686 \pm 0.037$ |
| RMSN-THLTS$^{(v)}$ | $0.695 \pm 0.008$ | $0.585 \pm 0.009$ | $0.573 \pm 0.017$ | $0.543 \pm 0.012$ | $0.561 \pm 0.016$ |
| RMSN-THLTS | $0.694 \pm 0.007$ | $0.578 \pm 0.007$ | $0.557 \pm 0.011$ | $0.522 \pm 0.018$ | $0.538 \pm 0.017$ |
| CRN | $0.621 \pm 0.005$ | $0.529 \pm 0.009$ | $0.551 \pm 0.017$ | $0.528 \pm 0.017$ | $0.586 \pm 0.019$ |
| CRN-THLTS$^{(v)}$ | $0.636 \pm 0.011$ | $0.515 \pm 0.007$ | $0.495 \pm 0.014$ | $0.457 \pm 0.015$ | $0.471 \pm 0.012$ |
| CRN-THLTS | $\mathbf{0.617 \pm 0.005}$ | $\mathbf{0.500 \pm 0.010}$ | $\mathbf{0.463 \pm 0.006}$ | $\mathbf{0.423 \pm 0.005}$ | $\mathbf{0.430 \pm 0.010}$ |
| CT | $0.668 \pm 0.005$ | $0.548 \pm 0.005$ | $0.521 \pm 0.005$ | $0.480 \pm 0.005$ | $0.480 \pm 0.005$ |
| CT-THLTS$^{(v)}$ | $0.885 \pm 0.005$ | $0.758 \pm 0.040$ | $0.738 \pm 0.014$ | $0.699 \pm 0.028$ | $0.707 \pm 0.029$ |
| CT-THLTS | $0.642 \pm 0.008$ | $0.520 \pm 0.007$ | $0.489 \pm 0.005$ | $0.453 \pm 0.007$ | $0.459 \pm 0.007$ |

As demonstrated in Table 1, MSM underperforms the other methods because it struggle with the complex variable relationship in the data generation. The models harnessing the power of deep learning (i.e. G-Net, RMSN, CRN, CT) achieve more precious forecast. When our proposed THLTS is applied in conjunction with the counterfactual outcome forecast models, their performance has been significantly enhanced. We find that larger strength of latent factors makes the forecast performance worse for all the models because the hidden heterogeneity problem is more severe. Under these scenarios, our method can leads to more remarkable performance enhancement, which shows the rationality of our method. Though THLTS$^{(v)}$ also offer advantages to these models, the performance improvement provided by THLTS$^{(v)}$ is inferior to that of THLTS.

We also conduct the experiments with different trajectory horizon $d$. Table 2 illustrates the experimental results. The advantages of THLTS over baseline methods become progressively larger as the trajectory horizon $d$ increases. This is because longer histories can facilitate more precise recovery of latent factors in our method.

We conduct experiments under the setting of time-varying latent factors $\mathbf{e}_t^{(i)}$ to justify our strategy of learning time-shared latent factors. We established the underlying latent factor to evolve around a given centroid. Formally, $\mathbf{e}_t^{(i)} = \bar{\mathbf{e}}^{(i)} + \mathcal{N}(0, \sigma_{vary})$, where $\sigma_{vary}$ controls the variation degree of latent factors across time steps. The results are presented in Figure 3, suggest that the performance enhancement provided by THLTS is overall more pronounced than that of THLTF$^{(v)}$. This shows the trade-off between model flexibility and forecast performance. When $\sigma_{vary}$ becomes larger, the time-varying part of latent factors surpasses the time-shared part, the margin becomes weaker. We also conduct experiments of time-varying latent factors without given centroid. The results also validate the effectiveness of our method. Additional experimental results can be found in Section A of Appendix.

## 5.3 SEMI-SYNTHETIC EXPERIMENT

**Dataset.** We used a semi-synthetic pipeline constructed by Melnychuk et al. (2022) using MIMIC-III (Johnson et al., 2016) to further verify the effectiveness of THLTS. Specifically, we extracted 1,000 patient trajectories with 25 vital covariates (including heart rate, sodium and red blood cell count) and 3 static covariates (gender, age and ethnicity) similarly to (Melnychuk et al., 2022). Partial outcome is generated by time-varying latent factors through an Gaussian process function, and the strength is controlled by a constant parameter $\alpha_g$, the details of semi-synthetic experiment is included in Section B of Appendix.

**Results.** To note that when the strength $\alpha_g$ becomes larger, the influence of latent factors on outcome generation across time steps also increases. As results shown in Table 3, after integration with our THLTS component, the counterfactual outcome forecast models achieve better forecast performance

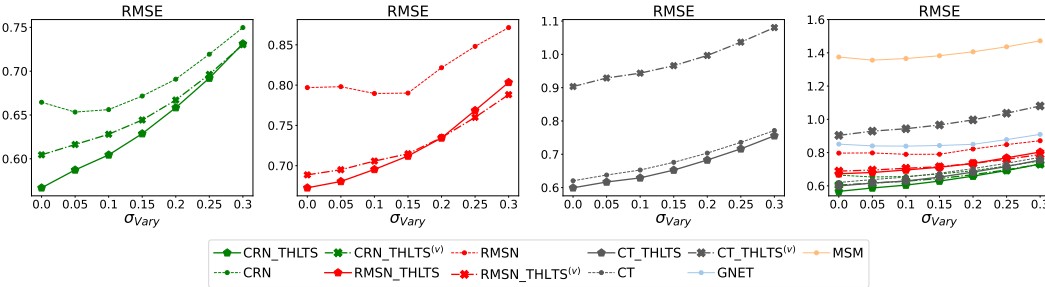

Figure 3: Performance of methods under the setting that the latent factors evolve around a given centroid across time steps. The x-axis represents different variation degree of latent factors (i.e. $\sigma_{vary}$). The rightest sub-figure shows the performance of all the methods. The leftest three sub-figures respectively shows the effect of THLTS on CRN, RMSN and CT.

Table 3: Results for semi-synthetic dataset, the average RMSE $\pm$ standard deviation is recorded for 10 repeated experiments.

| | Vary the heterogeneity strength $\alpha_g$ | | | | |
|---|---|---|---|---|---|
| $\alpha_g$ | $\alpha_g = 0.5$ | $\alpha_g = 1.0$ | $\alpha_g = 1.5$ | $\alpha_g = 2.0$ | $\alpha_g = 2.5$ |
| MSM | $1.127 \pm 0.000$ | $1.027 \pm 0.000$ | $1.118 \pm 0.000$ | $1.033 \pm 0.000$ | $0.762 \pm 0.000$ |
| G-Net | $0.410 \pm 0.050$ | $0.392 \pm 0.048$ | $0.397 \pm 0.046$ | $0.391 \pm 0.035$ | $0.396 \pm 0.037$ |
| RMSN | $0.256 \pm 0.006$ | $0.270 \pm 0.004$ | $0.268 \pm 0.006$ | $0.266 \pm 0.005$ | $0.271 \pm 0.004$ |
| RMSN-THLTS[v] | $0.252 \pm 0.008$ | $0.267 \pm 0.005$ | $0.265 \pm 0.006$ | $0.263 \pm 0.004$ | $0.269 \pm 0.004$ |
| RMSN-THLTS | $0.246 \pm 0.008$ | $0.252 \pm 0.004$ | $0.253 \pm 0.005$ | $0.255 \pm 0.005$ | $0.261 \pm 0.004$ |
| CRN | $0.254 \pm 0.018$ | $0.268 \pm 0.018$ | $0.287 \pm 0.015$ | $0.294 \pm 0.013$ | $0.300 \pm 0.013$ |
| CRN-THLTS[v] | $0.256 \pm 0.024$ | $0.265 \pm 0.020$ | $0.284 \pm 0.020$ | $0.291 \pm 0.020$ | $0.296 \pm 0.021$ |
| CRN-THLTS | $\mathbf{0.221 \pm 0.004}$ | $0.267 \pm 0.005$ | $0.264 \pm 0.006$ | $0.276 \pm 0.006$ | $0.282 \pm 0.006$ |
| CT | $0.263 \pm 0.008$ | $0.267 \pm 0.011$ | $0.268 \pm 0.009$ | $0.264 \pm 0.007$ | $0.263 \pm 0.007$ |
| CT-THLTS[v] | $0.259 \pm 0.010$ | $0.261 \pm 0.007$ | $0.261 \pm 0.005$ | $0.259 \pm 0.005$ | $0.257 \pm 0.005$ |
| CT-THLTS | $0.239 \pm 0.006$ | $\mathbf{0.240 \pm 0.006}$ | $\mathbf{0.246 \pm 0.005}$ | $\mathbf{0.247 \pm 0.003}$ | $\mathbf{0.247 \pm 0.003}$ |

than the original model. Compared to the version of capturing the time varying dynamics of latent factors THLTS[v], our proposed THLTS aims to capture the time-shared part, which acts as a regularizer constraining the learning of latent factors, and further improve the forecast performance. In summary, the semi-synthetic experiments on the MIMIC-III datasets confirm the effectiveness of our method.

## 6 CONCLUSION

In this paper, we studied the hidden heterogeneity problem of counterfactual outcome forecast in longitudinal setting. Neglecting the latent factors unobserved in the histories can degrade the prediction performance of counterfactual outcome forecast model. To tackle this problem, we propose Time-shared Heterogeneity Learning from Time Series (THLTS) method to uncover the time-shared part of latent factors and augment the forecast model with the learned latent representation. Owing to the flexibility of our method, it can be combined with arbitrary model backbones. Extensive experimental results indicate the effectiveness of our method on improving the performance of mainstream counterfactual outcome forecast models.

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

## A  ADDITIONAL EXPERIMENTS

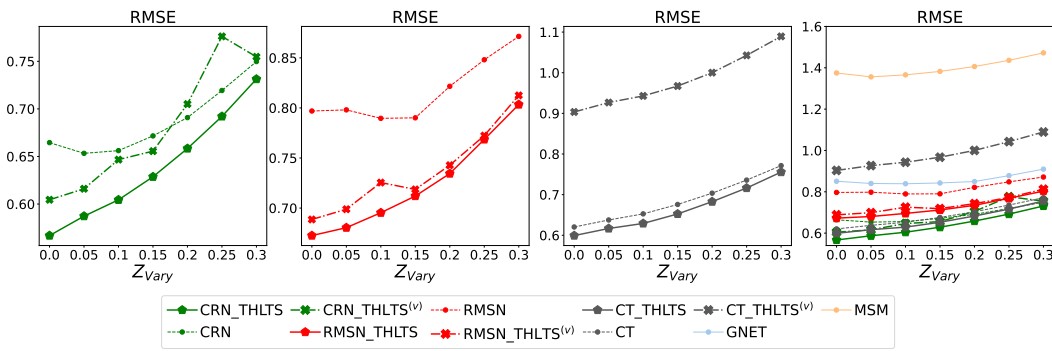

Figure 4: Performance of methods under the setting that the latent factors evolve without a given centroid. The x-axis represents different variation degree of latent factors (i.e. $\sigma_{vary}$). The rightest sub-figure shows the performance of all the methods. The leftest three sub-figures respectively shows the effect of THLTS on CRN, RMSN and CT.

**The performance of THLTS when latent factor evolves without a given centroid:** We consider a different scenario that latent factors evolves based on the value of the previous time step, formally $\mathbf{e}_t^{(i)} = \mathbf{e}_{t-1}^{(i)} + \mathcal{N}(0, \sigma_{vary})$, where $\sigma_{vary}$ is a constant that controls the variation degree of latent factor across time steps. We set $d = 20$ and $m = 1.5$. The results are shown in Figure 4, the trends are similar to the results in main paper.

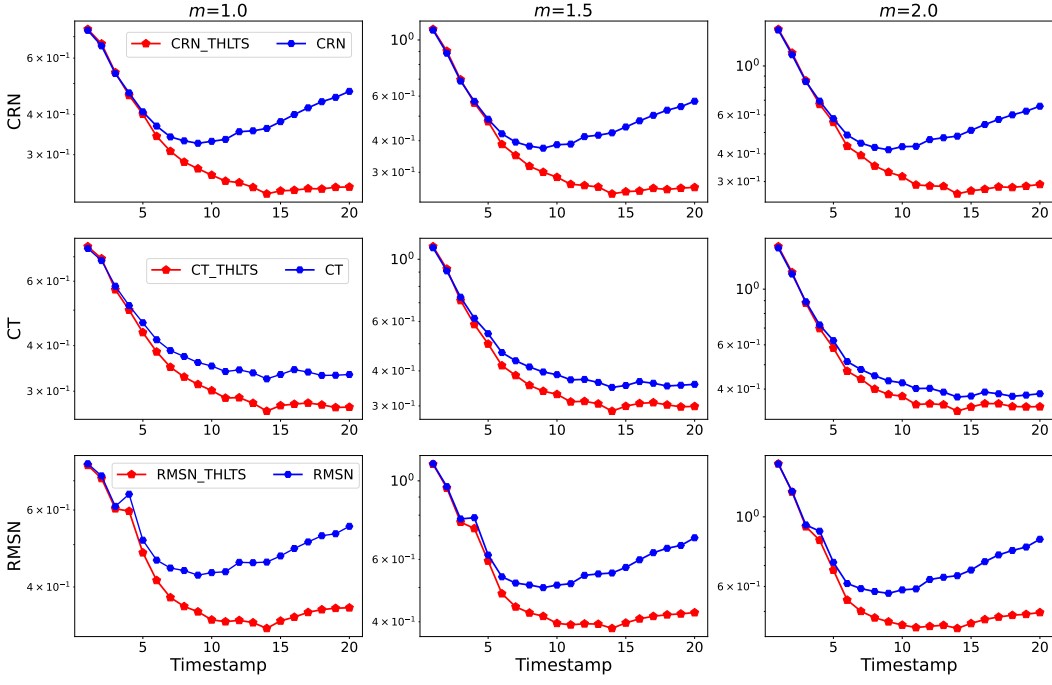

Figure 5: Counterfactual outcome forecast performance at each timestamp $t$.

**The performance of THLTS for each timestamp:** We further illustrate the performance of THLTS to estimate the counterfactual outcome at each time step $t$. Figure 5 depicts the results, the advantages of THLTS becomes progressively more prominent when the index of time step $t \geq 5$, by the virtue of more precise recovery of latent factors with longer histories data.

# B  EXPERIMENTAL DETAILS

## B.1  SEMI-SYNTHETIC EXPERIMENTS

We used the semi-synthetic dataset built from MIMIC-IIIJohnson et al. (2016) followed the pipeline introduced in Melnychuk et al. (2022); Schulam & Saria (2017). Specifically, we extracted 1,000 patient trajectories with time horizon $T^{(i)} = 20$. The covariates is composed of 25 time-varying patient signs (including heart rate, sodium and red blood cell count) and 3 static patients' information (gender, age and ethnicity).

Based on the extracted patient trajectories, there are three binary treatment and two continuous outcome simulated at each time step. The treatment assignment probability $p_{\mathbf{A}_t^l}$ is confounded by the patient covariates and previous outcome as follows:

$$p_{\mathbf{A}_t^l} = \sigma(\gamma_{\mathbf{A}}^l \bar{\mathbf{A}}_{T_l}(\bar{\mathbf{Y}}_{t-1} + \gamma_{\mathbf{X}}^l f_Y^l(\mathbf{X}_t) + b_l)), \tag{11}$$

where $\gamma_X^l, \gamma_A^l$ are constant parameters that control the confounding strength of treatment $\mathbf{A}_l$, $\sigma(\cdot)$ is a sigmoid function, $b_l$ is the constant bias for each treatment, then each binary treatment $A_t$ is sample from Bernoulli Distribution with parameter $\mathbf{A}_t^l$. The untreated outcome is then generated by the combination of endogenous and exogenous parts:

$$\mathbf{Y}_{untreated,t}^{(i),j} = \underbrace{\alpha_S^j \text{B-Spline}(t) + \alpha_g^j g^{j,(i)}(t)}_{\text{endogenous}} + \underbrace{\alpha_f^j f_Z^j(\mathbf{X}_\mathbf{t}^{(\mathbf{i})})}_{\text{exogenous}} + \epsilon_t, \tag{12}$$

where $\alpha_S^j, \alpha_g^j, \alpha_f^j$ are constant parameter and $\epsilon_t$ is sampled from $\mathcal{N}(0, 0.005^2)$. Here we focus on $g^{j,(i)}(t)$, which is generated by independently for each patient from Gaussian process with Matérn Kernel, which is equivalent to the time-varying latent factors discussed in this paper. $\alpha_g^j$ is a constant parameter controls the contribution of it.

We then generate treated outcome as following:

$$\mathbf{Y}_{treated,t}^{(i),j} = \mathbf{Y}_{untreated,t}^{(i),j} + \sum_{i=t-ws}^{t} \frac{min_{l=0,1,2}\mathbb{1}[\mathbf{A}_i^l = 1]p_{\mathbf{A}_i^l}\beta_{l,j}}{(w^l - i)^{0.5}} \cdot \alpha_g^j g^{j,(i)}, \tag{13}$$

where the individual latent factors also impact the patients' response of a given treatment.

## B.2  COMPUTATION RESOURCE

We conduct our experiments on a Linux server, where the operation system is 18.04.1-Ubuntu. There are 8 NVIDIA GeForce RTX 3090 GPUS on this server. However, we only use one GPU. The internal memory is 504GB. Each run of our experiments cost around 10 minutes.

# C  PROOF

**Proposition C.1.** *(Restated) Assuming the function of prediction model $g$ is $\beta$-Lipschitz on $\mathbf{e}$, formally $|g(\phi(\mathbf{H}), \mathbf{e}, \mathbf{a}) - g(\phi(\mathbf{H}), \mathbf{e}', \mathbf{a})| \le \beta \cdot ||\mathbf{e} - \mathbf{e}'||_2$, then the total increased error across time induced by substitute time-varying $\mathbf{e}_t$ with constant $\bar{\mathbf{e}}$ can be characterized as following:*

$$\sum_{t=1}^{T^{(i)}} (g(\phi(\mathbf{H}_t), \bar{\mathbf{e}}, \mathbf{a}_t^c) - \mathbf{Y}_t(\mathbf{a}_t^c))^2 \le 2 \sum_{t=1}^{T^{(i)}} (g(\phi(\mathbf{H}_t), \mathbf{e}_t, \mathbf{a}_t^c) - \mathbf{Y}_t(\mathbf{a}_t^c))^2 + 2\beta^2 \cdot \sum_{t=1}^{T^{(i)}} ||\mathbf{e}_t - \bar{\mathbf{e}}||_2^2 \tag{14}$$

*Proof.* For arbitrary time step $t \in \{1, 2, 3, ..., T^{(i)}\}$, we have

$$
\begin{aligned}
(g(\phi(\mathbf{H}_t), \bar{\mathbf{e}}, \mathbf{a}_t^c) - \mathbf{Y}_t(\mathbf{a}_t^c))^2 &= (g(\phi(\mathbf{H}_t), \bar{\mathbf{e}}, \mathbf{a}_t^c) - g(\phi(\mathbf{H}_t), \mathbf{e}_t, \mathbf{a}_t^c) + g(\phi(\mathbf{H}_t), \mathbf{e}_t, \mathbf{a}_t^c) - \mathbf{Y}_t(\mathbf{a}_t^c))^2 \\
&\le 2 \cdot (g(\phi(\mathbf{H}_t), \bar{\mathbf{e}}, \mathbf{a}_t^c) - g(\phi(\mathbf{H}_t), \mathbf{e}_t, \mathbf{a}_t^c))^2 \\
&+ 2 \cdot (g(\phi(\mathbf{H}_t), \mathbf{e}_t, \mathbf{a}_t^c) - \mathbf{Y}_t(\mathbf{a}_t^c))^2 \\
&= 2 \cdot (g(\phi(\mathbf{H}_t), \bar{\mathbf{e}}, \mathbf{a}_t^c) - g(\phi(\mathbf{H}_t), \mathbf{e}_t, \mathbf{a}_t^c))^2 + 2 \cdot (\beta ||\mathbf{e}_t - \bar{\mathbf{e}}||_2)^2
\end{aligned}
$$

By taking the sum of inequality for $t \in \{1, 2, 3, ..., T^{(i)}\}$, we obtain

$$\sum_{t=1}^{T^{(i)}} (g(\phi(\mathbf{H}_t), \bar{\mathbf{e}}, \mathbf{a}_t^c) - \mathbf{Y}_t(\mathbf{a}_t^c))^2 \leq 2 \sum_{t=1}^{T^{(i)}} (g(\phi(\mathbf{H}_t), \mathbf{e}_t, \mathbf{a}_t^c) - \mathbf{Y}_t(\mathbf{a}_t^c))^2 + 2\beta^2 \cdot \sum_{t=1}^{T^{(i)}} ||\mathbf{e}_t - \bar{\mathbf{e}}||_2^2$$

□

**Proposition C.2.** *(Restated) We assume the following conditions are satisfied:*

1. *The latent factors are generated by $\mathbf{e}_t = \bar{\mathbf{e}} + \eta_t$, where $\eta_t \sim p(\eta)$ is a noise term with zero mean.*

2. *The outcome distribution $p(\mathbf{y}_t|\mathbf{H}_t, \mathbf{a}, \bar{\mathbf{e}}) = \int_\eta p(\mathbf{y}_t|\mathbf{H}_t, \mathbf{a}, \mathbf{e} = \bar{\mathbf{e}} + \eta) \cdot p(\eta)d\eta$ is in the function family of decoder $p_\rho(\mathbf{y}_t|\mathbf{H}_t, \mathbf{a}, \bar{\mathbf{e}})$.*

3. *The posterior distribution $p(\bar{\mathbf{e}}|\mathbf{H}_t, \mathbf{a}_t, \mathbf{y}_t)$ is in the function family of encoder $q_\varphi(\bar{\mathbf{e}}|\mathbf{H}_t, \mathbf{a}_t, \mathbf{y}_t)$.*

*Then there is an optimal solution for maximizing the evidence lower bound (ELBO) of variational autoencoders, which characterizes the underlying data generation process:*

$$q_\varphi(\bar{\mathbf{e}}|\mathbf{H}_t, \mathbf{a}_t, \mathbf{y}_t) = p(\bar{\mathbf{e}}|\mathbf{H}_t, \mathbf{a}_t, \mathbf{y}_t), \quad p_\rho(\mathbf{y}|\mathbf{H}_t, \mathbf{a}, \bar{\mathbf{e}}) = p(\mathbf{y}|\mathbf{H}_t, \mathbf{a}, \bar{\mathbf{e}}), \quad p_\rho(\bar{\mathbf{e}}|\mathbf{H}_t) = p(\bar{\mathbf{e}}|\mathbf{H}_t),$$

*where the ELBO is defined as:*

$$\sum_{i=1}^n \sum_{j=1}^{T^{(i)}} \left( \mathbb{E}_{\bar{\mathbf{e}} \sim q_\varphi(\bar{\mathbf{e}}|\mathbf{H}_j^{(i)}, \mathbf{a}_j^{(i)}, \mathbf{y}_j^{(i)})} \left[ \log p_\rho(\mathbf{y}_j^{(i)}|\mathbf{H}_j^{(i)}, \mathbf{a}_j^{(i)}, \bar{\mathbf{e}}) \right] + D_{KL}(p_\rho(\bar{\mathbf{e}}|\mathbf{H}_j^{(i)})|q_\varphi(\bar{\mathbf{e}}|\mathbf{H}_j^{(i)}, \mathbf{a}_j^{(i)}, \mathbf{y}_j^{(i)})) \right)$$

*Proof.* We can decompose the joint distribution of outcome sequence as

$$\log p_\rho(\mathbf{Y}_{1:t}|\mathbf{X}_{1:t}, \mathbf{A}_{1:t}) = \sum_{i=1}^t \log p_\rho(\mathbf{y}_i|\mathbf{X}_{1:i}, \mathbf{A}_{1:i}, \mathbf{Y}_{1:i-1}) = \sum_{i=1}^t \log p_\rho(\mathbf{y}_i|\mathbf{H}_i, \mathbf{a}_i)$$

Specifically,

$$
\begin{aligned}
&\log p_\rho(\mathbf{y}_i|\mathbf{H}_i, \mathbf{a}_i) \\
=\ & \mathbb{E}_{\bar{\mathbf{e}} \sim q_\varphi(\bar{\mathbf{e}}|\mathbf{H}_i, \mathbf{a}_i, \mathbf{y}_i)}[\log p_\rho(\mathbf{y}_i|\mathbf{H}_i, \mathbf{a}_{1:i})] \\
=\ & \mathbb{E}_{\bar{\mathbf{e}} \sim q_\varphi(\bar{\mathbf{e}}|\mathbf{H}_i, \mathbf{a}_i, \mathbf{y}_i)} \left[ \log \frac{p_\rho(\bar{\mathbf{e}}, \mathbf{Y}_{1:t}|\mathbf{X}_{1:t}, \mathbf{A}_{1:t})}{p_\rho(\bar{\mathbf{e}}|\mathbf{X}_{1:t}, \mathbf{A}_{1:t}, \mathbf{Y}_{1:t})} \right] & (15) \\
=\ & \mathbb{E}_{\bar{\mathbf{e}} \sim q_\varphi(\bar{\mathbf{e}}|\mathbf{H}_i, \mathbf{a}_i, \mathbf{y}_i)} \left[ \log \frac{p_\rho(\bar{\mathbf{e}}, \mathbf{y}_i|\mathbf{H}_i, \mathbf{a}_i)}{q_\varphi(\bar{\mathbf{e}}|\mathbf{H}_i, \mathbf{a}_i, \mathbf{y}_i)} \cdot \frac{q_\varphi(\bar{\mathbf{e}}|\mathbf{H}_i, \mathbf{a}_i, \mathbf{y}_i)}{p_\rho(\bar{\mathbf{e}}|\mathbf{H}_i, \mathbf{a}_i, \mathbf{y}_i)} \right] & (16) \\
=\ & \mathbb{E}_{\bar{\mathbf{e}} \sim q_\varphi(\bar{\mathbf{e}}|\mathbf{H}_i, \mathbf{a}_i, \mathbf{y}_i)} \left[ \log \frac{p_\rho(\bar{\mathbf{e}}, \mathbf{y}_i|\mathbf{H}_i, \mathbf{a}_i)}{q_\varphi(\bar{\mathbf{e}}|\mathbf{H}_i, \mathbf{a}_i, \mathbf{y}_i)} \right] & (17) \\
& +\ D_{KL}(q_\varphi(\bar{\mathbf{e}}|\mathbf{H}_i, \mathbf{a}_i, \mathbf{y}_i)|p_\rho(\bar{\mathbf{e}}|\mathbf{H}_i, \mathbf{a}_i, \mathbf{y}_i)) & (18) \\
=\ & \mathbb{E}_{\bar{\mathbf{e}} \sim q_\varphi(\bar{\mathbf{e}}|\mathbf{H}_i, \mathbf{a}_i, \mathbf{y}_i)} \left[ \log \frac{p_\rho(\mathbf{y}_i|\mathbf{H}_i, \mathbf{a}_i, \bar{\mathbf{e}})}{q_\varphi(\bar{\mathbf{e}}|\mathbf{H}_i, \mathbf{a}_i, \mathbf{y}_i)} + \log p_\rho(\bar{\mathbf{e}}|\mathbf{H}_i, \mathbf{a}_i) \right] & (19) \\
& +\ D_{KL}(q_\varphi(\bar{\mathbf{e}}|\mathbf{H}_i, \mathbf{a}_i, \mathbf{y}_i)|p_\rho(\bar{\mathbf{e}}|\mathbf{H}_i, \mathbf{a}_i, \mathbf{y}_i)). & (20)
\end{aligned}
$$

Since the time-shared latent factors $\bar{\mathbf{e}}$ is independently of $\mathbf{a}$ given the histories $\mathbf{H}$, we have $p_\rho(\bar{\mathbf{e}}|\mathbf{H}_i, \mathbf{a}_i) = p_\rho(\bar{\mathbf{e}}|\mathbf{H}_i)$. Then we have

$$
\begin{aligned}
&\log p_\rho(\mathbf{y}_i|\mathbf{H}_i, \mathbf{a}_i) \\
=\ & \mathbb{E}_{\bar{\mathbf{e}} \sim q_\varphi(\bar{\mathbf{e}}|\mathbf{H}_i, \mathbf{a}_i, \mathbf{y}_i)} [\log p_\rho(\mathbf{y}_i|\mathbf{H}_i, \mathbf{a}_i, \bar{\mathbf{e}})] \\
& +\ D_{KL}(p_\rho(\bar{\mathbf{e}}|\mathbf{H}_i)|q_\varphi(\bar{\mathbf{e}}|\mathbf{H}_i, \mathbf{a}_i, \mathbf{y}_i)) & (21) \\
& +\ D_{KL}(q_\varphi(\bar{\mathbf{e}}|\mathbf{H}_i, \mathbf{a}_i, \mathbf{y}_i)|p_\rho(\bar{\mathbf{e}}|\mathbf{H}_i, \mathbf{a}_i, \mathbf{y}_i))
\end{aligned}
$$

We take the sum of equations above for all the training samples and time steps, and obtain

$$\sum_{i=1}^{n} \log p_\rho(\mathbf{Y}_{1:T^{(i)}}^{(i)} | \mathbf{X}_{1:T^{(i)}}^{(i)}, \mathbf{A}_{1:T^{(i)}}^{(i)})$$

$$- \sum_{i=1}^{n} \sum_{j=1}^{T^{(i)}} D_{KL}(q_\varphi(\bar{\mathbf{e}} | \mathbf{H}_j^{(i)}, \mathbf{a}_j^{(i)}, \mathbf{y}_j^{(i)}) | p_\rho(\bar{\mathbf{e}} | \mathbf{H}_j^{(i)}, \mathbf{a}_j^{(i)}, \mathbf{y}_j^{(i)}))$$

$$= \sum_{i=1}^{n} \sum_{j=1}^{T^{(i)}} \mathbb{E}_{\bar{\mathbf{e}} \sim q_\varphi(\bar{\mathbf{e}} | \mathbf{H}_j^{(i)}, \mathbf{a}_j^{(i)}, \mathbf{y}_j^{(i)})} \left[ \log p_\rho(\mathbf{y}_j^{(i)} | \mathbf{H}_j^{(i)}, \mathbf{a}_j^{(i)}, \bar{\mathbf{e}}) \right]$$

$$+ \sum_{i=1}^{n} \sum_{j=1}^{T^{(i)}} D_{KL}(p_\rho(\bar{\mathbf{e}} | \mathbf{H}_j^{(i)}) | q_\varphi(\bar{\mathbf{e}} | \mathbf{H}_j^{(i)}, \mathbf{a}_j^{(i)}, \mathbf{y}_j^{(i)})) \tag{22}$$

From the equations above, we can observe that when the following conditions are satisfied,

$$q_\varphi(\bar{\mathbf{e}} | \mathbf{H}_t, \mathbf{a}_t, \mathbf{y}_t) = p(\bar{\mathbf{e}} | \mathbf{H}_t, \mathbf{a}_t, \mathbf{y}_t), \quad p_\rho(\mathbf{y} | \mathbf{H}_t, \mathbf{a}, \bar{\mathbf{e}}) = p(\mathbf{y} | \mathbf{H}_t, \mathbf{a}, \bar{\mathbf{e}}), \quad p_\rho(\bar{\mathbf{e}} | \mathbf{H}_t) = p(\bar{\mathbf{e}} | \mathbf{H}_t).$$

We have

$$p_\rho(\bar{\mathbf{e}} | \mathbf{H}_t, \mathbf{a}_t, \mathbf{y}_t) = p(\bar{\mathbf{e}} | \mathbf{H}_t, \mathbf{a}_t, \mathbf{y}_t) = q_\varphi(\bar{\mathbf{e}} | \mathbf{H}_t, \mathbf{a}_t, \mathbf{y}_t)$$

$$\sum_{i=1}^{n} \sum_{j=1}^{T^{(i)}} D_{KL}(q_\varphi(\bar{\mathbf{e}} | \mathbf{H}_j^{(i)}, \mathbf{a}_j^{(i)}, \mathbf{y}_j^{(i)}) | p_\rho(\bar{\mathbf{e}} | \mathbf{H}_j^{(i)}, \mathbf{a}_j^{(i)}, \mathbf{y}_j^{(i)})) = 0$$

Therefore, the r.h.s of Equation 22, which is the ELBO of our variational autoencoders, becomes $\sum_{i=1}^{n} \log p(\mathbf{Y}_{1:T^{(i)}}^{(i)} | \mathbf{X}_{1:T^{(i)}}^{(i)}, \mathbf{A}_{1:T^{(i)}}^{(i)})$ and reaches the maximal value. $\qquad\square$

## D  LIMITATIONS

In this paper, we only theoretically analyze the setting where the latent factors are generated by a specific process $\mathbf{e}_t = \bar{\mathbf{e}} + \eta_t$, though the data generation process violating this assumption is also examined in the experiments. We leave the analysis of more complex scenarios to future works.

Due to the intrinsic challenges in evaluating counterfactual prediction, we only conduct experiments on the synthetic datasets and semi-synthetic datasets. Evaluation with expensive real-world experiments should also be considered.

## E  SYMBOL SUMMARY

We summarize the symbols and corresponding definitions in Table 4.

## F  BACKGROUND INFORMATION OF VAE

Generally, variational autoencoder is a framework that assume the observation $\mathbf{o}$ is generated from latent factors $\mathbf{z}$ and conditions $\mathbf{c}$ (if exists) and learn a variational approximation $q_\varphi(\mathbf{z} | \mathbf{o}, \mathbf{c})$ to substitute the true posterior distribution $p(\mathbf{z} | \mathbf{o}, \mathbf{c})$. The encoder outputting this approximated posterior is trained with decoder $p_\rho(\mathbf{o} | \mathbf{z}, \mathbf{c})$ and conditional prior component $p_\varphi(\mathbf{z} | \mathbf{c})$ to maximize a lower bound of the log-likelihood of the observed data $p(\mathbf{o} | \mathbf{c})$. Formally, the evidence lower bound (ELBO) is defined as:

$$\mathbb{E}_\mathcal{D}[\log p(\mathbf{o} | \mathbf{c})] \geq \mathcal{L}(\varphi, \rho) = \mathbb{E}_\mathcal{D}[\mathbb{E}_{q_\varphi(\mathbf{z} | \mathbf{o}, \mathbf{c})}[\log p_\rho(\mathbf{o} | \mathbf{z}, \mathbf{c}) + \log p_\varphi(\mathbf{z} | \mathbf{c}) - \log q_\varphi(\mathbf{z} | \mathbf{o}, \mathbf{c})]]$$

$$= \mathbb{E}_\mathcal{D}[\mathbb{E}_{q_\varphi(\mathbf{z} | \mathbf{o}, \mathbf{c})}[\log p_\rho(\mathbf{o} | \mathbf{z}, \mathbf{c})] - D_{KL}(\log q_\varphi(\mathbf{z} | \mathbf{o}, \mathbf{c}) | \log p_\varphi(\mathbf{z} | \mathbf{c}))]$$

Usually, the outputted distributions of these components are defined as Gaussian distribution with parameterized expectation and variance. After training the models, we can use reparameterization

Table 4: The symbols used in the paper and corresponding definitions

| Symbol | Definition |
| --- | --- |
| $\mathbf{x}_t^{(i)} \in \mathcal{X} \subset \mathbb{R}^{d_x}$ | Covariate of the $i^{th}$ sample at the $t^{th}$ time step |
| $\mathbf{a}_t^{(i)} \in \mathcal{A}$ | Treatment of the $i^{th}$ sample at the $t^{th}$ time step |
| $\mathbf{y}_t^{(i)} \in \mathbb{R}$ | Outcome of the $i^{th}$ sample at the $t^{th}$ time step |
| $T_t^{(i)} \in \mathbb{N}$ | The observed history length of the $i^{th}$ sample |
| $\mathbf{X}_{t:t+\tau}^{(i)}$ | The observed covariate history of the $i^{th}$ sample |
| $\mathbf{A}_{t:t+\tau}^{(i)}$ | The observed treatment history of the $i^{th}$ sample |
| $\mathbf{Y}_{t:t+\tau}^{(i)}$ | The observed outcome history of the $i^{th}$ sample |
| $\mathbf{H}_t^{(i)}$ | The observed history of the $i^{th}$ sample |
| $\phi(\cdot)$ | The sequence model learning the representation of history |
| $\mathbf{e}_t^{(i)}$ | The latent factors of the $i^{th}$ sample at the $t^{th}$ time step |
| $\bar{\mathbf{e}}^{(i)}$ | The time-shared latent factors of the $i^{th}$ sample |
| $q_\varphi(\cdot)$ | The encoder outputting the posterior of latent factors |
| $p_\rho(\cdot)$ | The decoder outputting the distribution of outcomes |
| $g_\rho(\cdot)$ | The predictive model (also a component in the decoder) |
| $m \in \mathbb{N}$ | The repeated sampling times for estimating counterfactual outcome |

trick to sample latent factors from $q_\varphi(\mathbf{z}|\mathbf{o}, \mathbf{c})$. Given the sampled latent factors, we can change the conditions to the target value and decode the desired new observations. Specifically, in this paper, the observed histories $\mathbf{H}_t$ and current treatment $\mathbf{a}_t$ play the role of conditions $\mathbf{c}$, observed outcome $\mathbf{y}_t$ act as the observation $\mathbf{o}$, and the time-shared latent factors substitute the latent factors $\mathbf{z}$. Therefore, the encoder outputs the variational approximation of posterior $q_\varphi(\bar{\mathbf{e}}|\mathbf{H}_t, \mathbf{a}_t, \mathbf{y}_t)$. We can sample time-shared latent factors given the histories, observed treatments and outcomes from $q_\varphi(\bar{\mathbf{e}}|\mathbf{H}_t, \mathbf{a}_t, \mathbf{y}_t)$, change the treatment (i.e. part of conditions) to be the counterfactual $\mathbf{a}_t^c$ (i.e. target value of conditions), and finally obtained the counterfactual outcome from the decoder $p_\rho(\mathbf{y}|\mathbf{H}_t, \mathbf{a}^c, \bar{\mathbf{e}})$.

# G  PSEUDO-CODE OF OUR METHOD

The working process of our algorithm can be found in Algorithm 1.

---

**Algorithm 1** Time-shared Heterogeneity Learning from Time Series (THLTS)

---

**Input:** Observational data $\{\{\mathbf{x}_t^{(i)}, \mathbf{a}_t^{(i)}, \mathbf{y}_t^{(i)}\}_{t=1}^{T^{(i)}}\}_{i=1}^n$, the histories of evaluated sample $\{\mathbf{x}_t, \mathbf{a}_t, \mathbf{y}_t\}_{t=1}^T$ and the counterfactual treatment $\mathbf{a}_{T+1}^c$ at the $(T+1)^{th}$ time step.

**Output:** Counterfactual outcome forecast $\hat{\mathbf{y}}$.

1: Train the backbone model for learning representation of histories $\phi$ and latent factor model, including encoder $q_\varphi(\cdot)$ and forecast model $g_\rho(\cdot)$.
2: Set $\mu_0 \leftarrow \mathbf{0}_e, \sigma_0 \leftarrow \mathbf{1}_e$.
3: **for** $k = 1, 2, ..., T$ **do**
4:     Update $\mu_k \leftarrow f_\varphi^\mu(\mathbf{y}_{k-1}, \mathbf{a}_{k-1}, \phi(\mathbf{H}_{k-1}), \mu^{k-1}, \sigma^{k-1})$
5:     Update $\sigma_k \leftarrow f_\varphi^\sigma(\mathbf{y}_{k-1}, \mathbf{a}_{k-1}, \phi(\mathbf{H}_{k-1}), \mu^{k-1}, \sigma^{k-1})$.
6: **end for**
7: Set $\hat{y} \leftarrow 0$. // Under out-of-sample setting.
8: **for** $k = 1, 2, ..., m$ **do**
9:     Sample $r \sim \mathcal{N}(0, \mathbf{I}_e)$.
10:     Compute $\bar{\mathbf{e}} \leftarrow \mu_T + r \odot \sigma_T$
11:     Update $\hat{y} \leftarrow \hat{y} + \frac{1}{m} \cdot g^\rho(\mathbf{a}_{T+1}^c, \phi(\mathbf{H}_{T+1}), \bar{\mathbf{e}})$.
12: **end for**
13: **return** Forecasted outcome $\hat{y}$.

---

