# OpenReview forum: "Learning Time-shared Hidden Heterogeneity for Counterfactual Outcome Forecast"
_ICLR.cc/2025/Conference — Submitted to ICLR 2025_

### Official Review · Reviewer_vdqa · 2024-10-27

**Soundness:** 4
**Presentation:** 3
**Contribution:** 2
**Rating:** 5
**Confidence:** 3

**Summary:**

This paper introduces Time-shared Heterogeneity Learning from Time Series (THLTS), a novel method for capturing hidden heterogeneity in longitudinal counterfactual outcome prediction. THLTS, designed as a flexible component that can be integrated with existing models, learns time-shared latent factors using VAE architecture.

**Strengths:**

1.The motivation is well-presented, which helps contextualize the problem being addressed.

2.The proof in Section 4.2 logically explains the motivation for employing a VAE architecture, making the rationale clear and reasonable.

3.Comprehensive experimental evaluation demonstrates performance improvements

**Weaknesses:**

1.The paper lacks novelty. The idea of recovering hidden factors has been widely explored in previous research. While learning the "TIME-SHARED" components is not as commonly discussed, but it is not significantly different from previous work, like Causal Effect Inference with Deep Latent-Variable Models" (Louizos et al., 2017),Causal Dynamic Variational Autoencoder for Counterfactual Regression in Longitudinal Data" (Bouchattaoui et al., 2023) and Factual Observation based Heterogeneity Learning for Counterfactual Prediction" (Zou et al., 2023).

2.While the paper mentions that "decision-making problems can span long periods of time," it does not introduce any specialized structures to capture unique features of long time series, such as periodicity or seasonality. For example, incorporating techniques like Fourier transforms for periodicity detection or wavelet transforms for handling multi-scale temporal structures could offer substantial improvements.

3.Despite claiming to address long-term time series forecasting, the paper only validates its method on notably short sequences (maximum 30 time steps).

**Questions:**

What are the unique challenges of addressing hidden heterogeneity across time?

---

> ### Author Response · Authors · 2024-11-28
>
> We are grateful for your insightful suggestions and constructive feedbacks. Below, we outlined the major contributions of our paper and address your concerns accordingly.
>
>
> It is noteworthy that our major contribution is not improving the capability of model architecture in dealing with long time-series, but leveraging the property of time series (treatment \& response interaction) data to recover hidden factors and address hidden heterogeneity under weaker assumptions than previous methods.
>
> Previous research[4] has revealed the importance of hidden heterogeneity induced by hidden factors. The previous methods utilize the information in high-dimensional treatments and proxies for the recovery of latent factors. However, in most real world scenarios, such as marketing promotions, medicine treatment effect, financial lending decision, treatments are mostly single-dimensional, rendering previous methods inapplicable. To adapt the idea of recovering hidden factors into more real applications, we proposed method that can capture hidden factors under single-dimensional treatment with time-series data, and we argue that time-series data such as e-commerce marketing, patient trajectories and financial records are ubiquitous in real world scenario. Specifically, we designed a novel mechanism that leverages the inherent advantage of possessing multiple outcome across time steps in time series to recover time-shared latent factors. Therefore, our contribution does not focus on improving the capability of model architecture in dealing with long time series (e.g. Causal Transformer leverage the powerful ability of Transformers in capturing the complex and long-range dependencies, to improve the model capability for counterfactual forecast). Instead, our paper focus on addressing the hidden heterogeneity problem brought by hidden factors.
> These two perspectives are **orthogonal** and **complementary**.
>
>
> To address the hidden heterogeneity problem, our proposed THLTS method acting as a flexible component instead of a new model architecture to recover the hidden factors, which can be combined with many off-the-shelf counterfactual forecast model. Therefore, in our implementations, we adopt the backbone model architecture same to the previous works, such as CRN, Causal Transformer. Further exploration on incorporating periodicity and seasonality to improve backbone model architecture can be left to future work.
>
> **Weakness 1**: The paper lacks novelty. The idea of recovering hidden factors has been widely explored in previous research.
>
> **Response**: We have illustrated the major contribution of our paper above, specifically, we consider the new opportunity and challenge brought by the temporal structure and design the corresponding solution.
> To be concrete, the hidden factors can vary across many time steps. Hence, the entire solution space of hidden factors can be extremely large which makes straightforwardly learning the hidden factors of each time step difficult. Moreover, the limited supervision information (i.e. the outcome variable is of few-dimension) of single time step further exacerbate the difficulty in learning hidden factors.
>
> To alleviate this issue, we propose a new strategy which leverages the multiple outcome across time steps to learn the time-shared latent factors exclusive to each sample and consequently constrain the over-flexible space of latent factor learning. Although it sacrifices the flexibility in
> modeling time-varying dynamics of latent factors, this design constrains the model flexibility and acts like a regularizer to improve forecast performance. Taking these into consideration, our work does not simply adopt the existing idea of recovering hidden factors, but give the design leveraging the exclusive problem property. Therefore, our contribution is sufficient.
>
> **Weakness 2:** It does not introduce any specialized structures to capture unique features of long time series
>
> **Response**: Thanks for your valuable suggestions. Since the major motivation of our paper is to resolve the hidden heterogeneity induced by hidden factors, we propose a flexible component THLTS that can be combined with off-the-shell counterfactual forecast models. The combined backbone use sequence models, such as LSTM and Transformers to capture the dependencies among time steps. Based on these, our THLTS method complement the design for temporal structure by setting the prior distribution at the $(t+1)^{th}$ step as the obtained posterior distribution at the $t^{th}$ step of the same sample according to the Bayes’ Theorem. The idea behind this design is that the observation and information of early stages can serves as the evidence for inference at the later stages in time series.

---

> > ### Author Response · Authors · 2024-11-28
> >
> > **Weakness 3**: The paper only validates its method on notably short sequences (maximum 30 time steps)
> >
> > **Response**: As we stated above, our main contribution is to resolve hidden heterogeneity problem brought by hidden factors, instead of proposing new model architectures to improve the capability in dealing with significantly long temporal structure. Therefore, the length of time series in our experiments is set to be comparable to that of previous works on counterfactual forecast in time series. The maximum sequence length in these works is between 20 and 60 [1,2,3], which is not significantly larger than ours.
> >
> > **Question 1**: What are the unique challenges of addressing hidden heterogeneity across time?
> >
> > **Response**: The unique challenge of addressing hidden heterogeneity is that the supervision information from single time step (i.e. outcome of single time step) is limited, and the solution space of all latent factors (the joint space of time-varying latent factors across time steps) can be extremely large. In contrast, in the previous works, this challenge is significantly weaker under static setting. On the one hand, the solution space of latent factors is small because it only considers the latent factors of one time step. On the other hand, the supervision information for recovering latent factors is sufficient. For example, in previous works [4], the latent variables are recovered from not only outcomes but also high-dimensional treatments and proxies. To overcome the challenge arising from time series data, in this paper, we propose a novel mechanism that leverage the inherent advantage of multiple outcomes across time steps to learns the time-shared part of latent factors and neglect the time-varying part so that the latent space for solution is significantly reduced. This design significantly improve the performance compared with the counterpart of directly learning time-varying latent factors.
> >
> > **Reference:**
> >
> > [1] Bryan Lim, Alaa Ahmed, and Mihaela van der Schaar. Forecasting treatment responses over time
> > using recurrent marginal structural networks. Advances in neural information processing systems,
> > 31, 2018.
> >
> > [2] Ioana Bica, Ahmed M Alaa, James Jordon, and Mihaela van der Schaar. Estimating counterfactual
> > treatment outcomes over time through adversarially balanced representations. arXiv preprint
> > arXiv:2002.04083, 2020b.
> >
> > [3] Valentyn Melnychuk, Dennis Frauen, and Stefan Feuerriegel. Causal transformer for estimating
> > counterfactual outcomes. In Proceedings of the 39th International Conference on Machine
> > Learning, volume 162 of Proceedings of Machine Learning Research, pp. 15293–15329. PMLR, 17–
> > 23 Jul 2022.
> >
> > [4] Hao Zou, Haotian Wang, Renzhe Xu, Bo Li, Jian Pei, Ye Jun Jian, and Peng Cui. Factual observation
> > based heterogeneity learning for counterfactual prediction. In Proceedings of the Second Conference on Causal Learning
> > and Reasoning, volume 213 of Proceedings of Machine Learning Research, pp. 350–370. PMLR,
> > 11–14 Apr 2023.

---

### Official Review · Reviewer_EH7E · 2024-11-01

**Soundness:** 3
**Presentation:** 2
**Contribution:** 2
**Rating:** 5
**Confidence:** 5

**Summary:**

The paper tackles the challenge of forecasting counterfactual outcomes in longitudinal settings. Previous methods using LSTM networks and transformers often neglect hidden heterogeneity caused by unobserved factors, which complicates predictions. The authors propose the Time-shared Heterogeneity Learning from Time Series method, which captures shared hidden factors using variational encoders. This approach enhances any counterfactual forecasting method and demonstrates improved performance in experiments with synthetic datasets.

**Strengths:**

1. Forecasing counterfactual prediction is highly applicable in real-world scenarios.
2. The time-shared heterogeneity based learning method is easy to implement with VAE.
3. This paper first utilizes longitudinal method to find the latent factor of each sample, which is intuitive.

**Weaknesses:**

1. In Proposition 4.1, it would be helpful for the authors to explain more about when the prediction model $g$ is Lipschitz with respect to $e$, as this is critical for ensuring the model's effectiveness in identifying the latent factor.
2. Since the latent factor is not directly observed, how can you guarantee that the latent factor identified by your method is the one you intend to find? It would be beneficial to provide some analysis regarding the identifiability of your method.
3. Why did you choose VAE to implement your method? Could other structures, such as deterministic models, serve as the backbone? If so, is it possible to test different models as backbones in the experimental section?
4. The compared baselines are not state-of-the-art methods. It would be better to select more recent methods as baselines to demonstrate the effectiveness of your approach, such as [1].



[1] Estimating Counterfactual Treatment Outcomes over Time through Adversarially Balanced Representations. ICLR 2020.

**Questions:**

See weakness

---

> ### Author Response · Authors · 2024-11-28
>
> We appreciate your valuable comments and constructive feedbacks on our manuscript.
>
> **Weakness 1**: In Proposition 4.1, it would be helpful for the authors to explain more about when the prediction model $g$
>  is Lipschitz with respect to $e$.
>
> **Response**:
> It is not a restricted hypothesis for the prediction module $g$ to be Lipschitz w.r.t latent factors $e$. The model $g$ is designed as a Multilayer Perceptron (MLP) comprised of fully-connected layers, and can be viewed as the composition of linear functions and non-linear activation functions $z_i=\sigma(W_iz_{i-1}+\mathbf{b}_i), 1\leq i \leq K$. The linear functions $Wz+\mathbf{b}$ are $\alpha$-Lipschitz continuous function w.r.t the input, where the constant $\alpha$ is the spectral norm of the
> weight matrix $\mathbf{W}$. The typical activation functions, including Sigmoid, ReLu and Tanh, are also Lipschitz continuous function, as the derivatives of them are bounded. For example, the Lipschitz constant of ReLu function is 1, and that of sigmoid function is $\frac{1}{4}$. Therefore, the composition of these Lipschitz continuous functions (i.e. the prediction model $g$) is also a Lipschitz continuous function.
>
> Moreover, the hypothesis of Lipschitz function is broadly adopted in other related works of causal inference [1,2,3], where the loss function is assumed to be Lipschitz w.r.t the input covariates.
>
> In summary, it is not a restricted hypothesis that the model $g$ is Lipschitz with respect to the part of input vector $e$.
>
>
> **Weakness 2**: It would be beneficial to provide some analysis regarding the identifiability of your method.
>
> **Response:** It has been demonstrated by the previous literature [4,5] in this community that the identifiability of latent factor recovery is difficult to be theoretically guaranteed without restricted assumption. However, it does not hinder the practical value of it which has been justified by these works. As a complement, we conduct an empirically analysis to show that the learned latent factors can indicate the true hidden factors.
>
> To be concrete, we train a predictor mapping the sampled latent factors to the true hidden factors. Lower prediction error implies that the learned latent factors is more closely related to the true hidden factors, which means higher degree of identifiablity. Specifically, we set the sequence length to 30 and employed the same hyperparameters to train our proposed model, which is based on the Causal Transformer architecture. We then derived the inferred latent factor $\bar{\mathbf{e}}^{(i)}$ at $t^{th}$ time stamp, meaning it relies solely on the information ($X_{1:t}^{(i)}, A_{1:t}^{(i)}, \mathbf{Y}_{1:t}^{(i)}$) available up to the
>  $t^{th}$ time stamp. To note that for the time stamp $t = 0$, the sampled latent factor is initialised as  $\bar{\mathbf{e}} ^{(i)} \sim \mathcal{N}(\mathbf{0}, \mathbf{I}_e)$ without any observation information. Subsequently, we utilized a Linear Regression model as the predictor backbone mapping the inferred $\bar{\mathbf{e}} ^{(i)}$ to the ground truth latent factor. The results are demonstrated in Table 1, the regression MSE decrease significantly from  $t=0$ to $t=15$, and reach a significantly low level after $t=15$. This empirically validates that the latent factors learned by our method is closely related to the true hidden factors that we intend to find. In Figure 5 of our original paper, we can observe a similar phenomenon that the advantages of THLTS over backbone models become progressively more prominent since $t > 5$, which can be explained by the more precise recovery of latent factor with longer histories data.
>
>
> Time step | 0 | 5|10|15|20|25|30
> ---------|----------|---------|---------|---------|---------|---------|---------
> MSE $\pm$ SD|1.9515 $\pm$ 0.048|0.4951 $\pm$ 0.027 |	0.1795 $\pm$ 	0.013 | 0.1229 $\pm$ 0.015|0.1113	$\pm$ 0.014  | 0.1128$\pm$ 0.014 |	0.1156 $\pm$ 0.020

---

> > ### Author Response · Authors · 2024-11-28
> >
> > **Weakness 3**: Why did you choose VAE to implement your method?
> >
> > **Response**: There are two primary reasons that we choose VAEs as a component in our proposed method. Firstly, it has the significant power in modelling the stochastic data generation process, particularly where variables are generated with exogenous uncertainty. This feature is crucial for our method that it allows us to effectively capture the probabilistic nature of the hidden factors. Secondly, the VAE makes substantially weaker assumptions about the data generating process and the structure of latent variables. Hence, it has the advantage in dealing with the various complex data scenarios.
> >
> > The deterministic models, such as normalizing flows and generative adversarial networks (GAN), are not appropriate candidates to serve as the backbone of our method.
> > The main reason is that the outcome variable is not determined by the pursued time-shared latent factor but also affected by the time-varying part of latent factors and exogenous noise. This is contradictory with the property of these models which characterize the deterministic relationship between latent factors and observations.Additionally, GANs, while effective in decoding latent factors to observations, lack the encoder component necessary for inferring latent factors from observations—a critical aspect of our method.
> >
> > **Weakness 4**: The compared baselines are not state-of-the-art methods. It would be better to select more recent methods as baselines to demonstrate the effectiveness of your approach, such as "Estimating Counterfactual Treatment Outcomes over Time through Adversarially Balanced Representations".
> >
> > **Response**: The CRN method you stated has been included as the baseline in the original version of our paper. The experiment section has covered the most representative and effective instances of counterfactual forecast methods in time series.  Furthermore, the most recent advance baseline in our paper is Causal Transformer that was publish at ICML 2022, which we believe is a SOTA approach that uses Transformer architecture.
> >
> >
> > **Reference:**
> >
> > [1] Shalit U, Johansson F D, Sontag D. Estimating individual treatment effect: generalization bounds and algorithms International conference on machine learning. PMLR, 2017: 3076-3085.
> >
> > [2] Assaad S, Zeng S, Tao C, et al. Counterfactual representation learning with balancing weights International Conference on Artificial Intelligence and Statistics. PMLR, 2021: 1972-1980.
> >
> > [3] Johansson F D, Kallus N, Shalit U, et al. Learning weighted representations for generalization across designs. arXiv preprint arXiv:1802.08598, 2018.
> >
> > [4] Louizos C, Shalit U, Mooij J M, et al. Causal effect inference with deep latent-variable models. Advances in neural information processing systems, 2017, 30.
> >
> > [5] Miao W, Hu W, Ogburn E L, et al. Identifying effects of multiple treatments in the presence of unmeasured confounding[J]. Journal of the American Statistical Association, 2023, 118(543): 1953-1967.

---

> > > ### Comment · Reviewer_EH7E · 2024-12-03
> > >
> > > Thanks for the authors' response, some of my concerns are solved. I must point out, the authors provide the rebuttal results out of the regular discussion time (Nov 27). Based on the overall judgement about this paper, the proposed method is not well motivated based on the existence of CRN [1] and Time Series Deconfounder [2], and the experimental visualization of the hidden factors is missing. I think this paper is not ready for publication in its current version, thus, I decide to maintain my score.
> > >
> > > **References:**
> > >
> > > [1] Bica, I., Alaa, A. M., Jordon, J., & van der Schaar, M. (2020). Estimating counterfactual treatment outcomes over time through adversarially balanced representations. arXiv preprint arXiv:2002.04083.
> > >
> > > [2] Bica, I., Alaa, A., & Van Der Schaar, M. (2020, November). Time series deconfounder: Estimating treatment effects over time in the presence of hidden confounders. In International conference on machine learning (pp. 884-895). PMLR.

---

> > > > ### Author Response · Authors · 2024-12-03
> > > >
> > > > Following the recent notification of an extension to the discussion period beyond the original deadline of November 27, we have decided to take additional time to further refine our responses and post them within the new discussion period. We believe this is in compliance with ICLR's regulations.
> > > >
> > > > We want to argue that the previous works including CRN and Time series Deconfounder can not negate our contributions. For CRN, it brought treatment-invariant representation learning method from domain adaptation to remove confounding bias in the data. For Time series Deconfounder, they borrow the idea of "The Blessing of multiple Cause" to recover the unobserved confounders and thereby remove the confounding bias. However, they neglect the hidden heterogeneity (i.e. the focus of our paper), the significance of which has been claimed in our Introduction. To address this problem, we leverage the inherent property of time series data and propose the method that fully utilize the outcome supervision across time steps. Additionally, compared to the Time Series Deconfounder, our proposed method can handle single-dimensional treatments, which broadens the applicability of hidden factor recovery to more scenarios.
> > > >
> > > > The validity of the learned latent factors have been sufficiently justified by the overall performance improvement in our experiments and the supplemented examination in rebuttal. The visualization of them can be a beneficial supplement. But we think it can not be a significant weakness for rejection.

---

### Official Review · Reviewer_h8m9 · 2024-11-03

**Soundness:** 2
**Presentation:** 3
**Contribution:** 2
**Rating:** 6
**Confidence:** 4

**Summary:**

This paper introduces a Time-Shared Heterogeneity Learning from Time Series (THLTS) approach for Counterfactual Outcome Forecasting, addressing the limitation in previous sequential models caused by insufficient consideration of hidden heterogeneity in sample outcomes caused by hidden factors. Extensive experiments demonstrate the effectiveness of THLTS, as well as its robustness in scenarios with unstable hidden factors or long sequence data.

**Strengths:**

1.This paper addresses the high relevance between decision-making tasks and temporal sequences, thoroughly analyzing the limitations of previous methods in modeling hidden factors beyond historical records. Leveraging a Variational Autoencoder (VAE), it creatively proposes a Time-Shared hidden factor learning approach to effectively bridge these gaps, demonstrating both originality and significance in the field.

2.This paper begins by presenting intuitive examples to illustrate how hidden factors can lead to different counterfactual outcomes for individuals with identical historical information. It then introduces the proposed THLTS method in a progressively detailed manner, followed by a rigorous theoretical derivation to analyze the validity of this hidden factor learning approach. Subsequently, the paper provides a detailed description of the three key components that enable THLTS, as well as its training and inference processes. The logic is coherent.

3.This paper conducts comprehensive experiment, validating the effectiveness of THLTS and its flexibility as a plugin of pioneering models, particularly under conditions of unstable hidden factors and long sequence data.

**Weaknesses:**

1.Clarity issues in some details. This is evident, on one hand, in the mismatch between figure legends and definitions in the text. For example, in the problem description, sample indices are indicated as superscripts, yet in Figure 1, sample indices for each latent factor are shown as subscripts, which potentially confuses them with time indices. On the other hand, some symbols lack explicit explanations; for instance, m in Equation (5), while seemingly representing the number of repetitions for sampling Time-Shared Hidden Factors and outcome prediction, would benefit from explicit clarification to enhance understanding.

2. Lacking sufficient background information. It particularly in explaining of VAE. For readers unfamiliar with variational inference, it may be challenging to understand how to model and sample the Time-Shared Hidden Factors.

3. The experiments are not sufficiently extensive. On one hand, the experimental data heavily relies on synthetic datasets, which significantly reduces the persuasiveness of the results and raises concerns about the practical applicability of the model. On the other hand, the choice of comparison baselines appears to lack novelty, as the most recent baseline, G-net, was proposed in 2020. Exploring and discussing more recent methods to highlight the contribution of this study would be beneficial.

**Questions:**

Q1: One argument in the paper posits that the modeling approach using Time-Shared Hidden Factors across all time steps for each sample is superior to previous methods that model hidden factors differently at each time step. This claim may appear somewhat counterintuitive and perplexing. Beyond the empirical conclusions drawn from experiments and the potential rationale of limited supervisory signals, is there a more comprehensive explanation or theoretical justification to alleviate this concern?

Q2: In the experiments, synthetic datasets were used. Firstly, does the synthetic method employed in the paper generate data that aligns with real-world distributions? Is this synthetic approach a standard in the field or a heuristic design by the authors? Furthermore, do the experimental results based on these datasets have practical significance? How is this demonstrated or substantiated?

Q3: There are some clarity-related concerns. Firstly, the learning of Time-Shared Hidden Factors is based on VAE, which is not reflected in the overall illustration in Figure 2. While appropriate simplification is essential, would incorporating the VAE structure into the diagram help readers better understand the model architecture? Additionally, in Section 4.3, could the mathematical description of L_t^((i)) be streamlined to aid readers in comprehending the model implementation? For instance, specifying that the KL divergence term involves the normal distributions corresponding to two contiguous time steps if correct.

---

> ### Author Response · Authors · 2024-11-28
>
> We appreciate your insightful comments and positive feedback. In response to your suggestions, we have made the revised version of our manuscript to enhance its comprehensiveness and clarity.
>
> **Weakness 1**: Clarity issues in some details
>
> **Response**: Thank you for pointing out the presentation issues of our paper. We have addressed these concerns in the revised manuscript to minimize potential confusion for readers.
>
> Specifically, we have revised Figure 1 and changed the sample indices for each latent factor to superscripts for better clarity.  Besides, we have added the description of $m$ in Line 348-349. To further ensure the clear definition of symbols, we have added a table summarizing all used symbols in Section E of the Appendix.
>
> **Weakness 2**: Insufficient background information
>
> **Response**:  We have supplemented the background information of VAE to provide readers with a clearer understanding of the underlying concepts. Hopefully this will make it easier for readers to grasp the model design and the pipeline of our proposed method. It can be found in the Section F of Appendix.
>
> **Weakness 3**: The experiments are not sufficiently extensive
>
> **Response**: Thank you for your feedback regarding the experiments. We will address your concerns as follows:
>
> ***Reliance on synthetic dataset***: Since counterfactual prediction aims to estimate the outcome of a unit under different treatments, which inherently requires groundtruth outcomes for all possible treatments, including the counterfactual treatments not observed in datasets. This makes it impractical to evaluate these methods in real-world scenarios [4]. For example, the commonly used dataset in temporal counterfactual learning literature is MIMIC-III, which records factual treatments and outcomes of patients. Specifically, the previous works in this field [1,2,3,4,5] mainly relies on the (semi-)synthetic datasets for empirical evaluation. In line with these works, we follow this established protocol to validate our proposed method. We use real-world features (collected patients status) to enhance the applicability and realism of our empirical results.
>
>
> ***Comparison baselines appears to lack novelty***: The most advanced baseline we have included in our original paper is Causal Transformer (it incorporates Transformer Architecture into Counterfactual Prediction task) and was published at ICML 2022. Additionally, we have incorporated the most recent baselines that leverage different causal learning mechanisms, such as G-Net (G-Computation, ML4H 2021), CRN (Invariant Learning, ICLR 2020), and RMSN (IPS-Weighted, NeurIPS 2018). These choices ensure that our experiments are both relevant and competitive within the current state of the field.
>
> **Question 1**: Is there a more comprehensive explanation or theoretical justification to the superiority of Time-Shared Hidden Factors?
>
> **Response**:
> On the one hand, the superior of modelling Time-Shared Hidden Factors is empirically justified in the experimental results. Specifically, the version of our method labeled THLTS$^{(v)}$ aims to capture the dynamic part of latent factors, which sets the prior distribution $\mathcal{N}(\mu^{pr}, \sigma^{pr})$ as a transformation of the posterior distribution $\mathcal{N}(\psi_{\mu}(\mu_{t-1}^{(i)}), Diag(\psi_{\sigma}(\sigma_{t-1}^{(i)})))$ instead of the original posterior distribution $\mathcal{N}(\mu_{t-1}^{(i)}, Diag(\sigma_{t-1}^{(i)}))$ of the previous time step. Our experimental results show that the THLTS model outperforms the THLTS$^{(v)}$ model, validating the effectiveness of learning time-shared latent factors.
>
>
> The rationale behind this improvement can be attributed to the model regularization effect, which constrains the flexibility of the model. The traditional practice in machine learning have revealed that the excessively flexible model can suffer from overfitting problem. To mitigate this issue, various regularizers have been proposed to reduce overfitting and enhance predictive performance. Inspired by these findings, we design the mechanism of learning time-shared latent factors which constrains the solution space of latent factors and play a similar role to regularizers in learning temporal counterfactual outcome with latent factors. The more rigorous theoretical analysis is attractive and requires substantial effort. We decide to leave it to future work.

---

> > ### Author Response · Authors · 2024-11-28
> >
> > **Question 3**: There are some clarity-related concerns
> >
> > **Response**: Thank you for your valuable suggestions. We have modified our diagram in Figure 2 to incorporate the VAE structure and demonstrate the architecture of our model. We also have adjusted the description in Section 4.3 and streamline $\mathcal{L}_t^{(i)}$ as you suggested to clarify the algorithms more clearly.
> >
> > **Reference:**
> >
> > [1] Mouad El Bouchattaoui, Myriam Tami, Benoit Lepetit, and Paul-Henry Cournède. Causal dynamic variational autoencoder for counterfactual regression in longitudinal data. arXiv preprint
> > arXiv:2310.10559, 2023.
> >
> > [2] Ioana Bica, Ahmed M Alaa, James Jordon, and Mihaela van der Schaar. Estimating counterfactual
> > treatment outcomes over time through adversarially balanced representations. arXiv preprint
> > arXiv:2002.04083, 2020b.
> >
> > [3] Bryan Lim, Alaa Ahmed, and Mihaela van der Schaar. Forecasting treatment responses over time
> > using recurrent marginal structural networks. Advances in neural information processing systems,
> > 31, 2018
> >
> > [4] Ioana Bica, James Jordon, and Mihaela van der Schaar. Estimating the effects of continuous-valued
> > interventions using generative adversarial networks. Advances in neural information processing
> > systems (NeurIPS), 2020c.
> >
> > [5] Hao Zou, Haotian Wang, Renzhe Xu, Bo Li, Jian Pei, Ye Jun Jian, and Peng Cui. Factual observation
> > based heterogeneity learning for counterfactual prediction. In Proceedings of the Second Conference on Causal Learning
> > and Reasoning, volume 213 of Proceedings of Machine Learning Research, pp. 350–370. PMLR,
> > 11–14 Apr 2023.
> >
> > [6] Ioana Bica, Ahmed Alaa, and Mihaela Van Der Schaar. Time series deconfounder: Estimating
> > treatment effects over time in the presence of hidden confounders. In International Conference
> > on Machine Learning, pages 884–895. PMLR, 2020.

---

### Author Response · Authors · 2024-12-03
**General responses for summary**

Dear Reviewers and Area Chair,

We sincerely appreciate the insightful reviews and the effort invested by both the reviewers and the AC. During the author-reviewer discussion period, we carefully considered the suggestions provided, addressed the questions, and revised the manuscript accordingly.


As noted by the reviewers, our paper has several promising strengths:

1) **Motivation and Relevance**: Our paper emphasizes the importance of hidden heterogeneity in decision-making under temporal sequences, with well-grounded motivation. (Reviewers h8m9, EH7E, vdqa)

2) **Theoretical Analysis**: Our rigorous theoretical analysis supports the rationality and validity of our proposed method. (Reviewers h8m9, vdqa)

3) **{Experimental Evaluation**: Comprehensive experiments provide strong evidence of the effectiveness of our approach. (Reviewers h8m9, vdqa)

Below, we address the key concerns raised by the reviewers:

1) **Validity and Novelty of Time-Shared Latent Factor Learning:**

The proposed strategy of learning time-shared latent factors leverages unique properties of time-series data to jointly infer latent factors across multiple outcomes over time. Unlike previous approaches, which rely on strong assumptions such as high-dimensional treatments or proxy variables, our method does not require such strong supervision.

2) **The use of (semi-)synthetic datasets**

Since counterfactual prediction aims to estimate the outcome of a unit under different treatments, which inherently requires groundtruth outcomes for all possible treatments. This makes it impractical to evaluate these methods in real-world scenarios. Thereby, the previous works in causality mainly relies on the (semi-)synthetic datasets for empirical evaluation. In line with these works, we follow this established protocol to validate our proposed method.

3) **Baselines in comparison**


Reviewer h8m9 identified G-net as the most advanced baseline, while Reviewer EH7E mentioned the omission of CRN. We clarify that our experiments include CRN, and the most advanced baseline utilized is Causal Transformer (published at ICML 2022), not G-net. We believe there may have been a misunderstanding regarding some experimental details.

We have also revised parts of the manuscript to enhance clarity and presentation, with the changes highlighted in red for easier reference. We hope our responses and the revisions adequately address all concerns. **If you have any further concerns or questions for our paper, we are more than willing to engage in a follow-up discussion with you!**

---

### Meta-Review · Area_Chair_UYkU · 2024-12-16

**Metareview:**

The paper introduces a method to perform counterfactual outcome prediction with hidden heterogeneity in a longitudinal setting. In terms of strengths, the reviewers appreciated the importance of the problem addressed by the authors and the presentation. In terms of weaknesses, the reviewers raised concerns regarding the significance of the technical contribution, the experimental setting and baselines used in the experimental evaluation, and the lack of theoretical justification for the method. Two of the reviewers were mildly negative in their overall evaluation of the paper and one of the reviewers was mildly positive. Based on the reviews and the rebuttal, I am unable to recommend acceptance -- one of the points that personally I find unconvincing is to rely on semi-synthetic experiments for the evaluation of a heuristic method for counterfactual inference. While it is true that related work on counterfactual inference resorts to semi-synthetic experiments, as the authors point out in their rebuttal, related work often include theoretical bounds and/or properties to ground proposed methodology.

**Additional Comments On Reviewer Discussion:**

The reviewers raised a number of concerns in their reviews, which the authors tried to address during the rebuttal period. However, the rebuttal did not persuade the reviewers to follow up or change their overall evaluation of the paper. As a result, I am unable to recommend acceptance.

---

### Decision · Program_Chairs · 2025-01-22

Reject